# SafeLab: An Interactive High-Fidelity Benchmark for Embodied Safety in Scientific Robotics

**Fengshuo Bai** [* 1 2 3] **Yufeng Li** [* 1 2 3] **Ruihai Wu** [* 4 5] **Peishuo Wang** [1] **Yuhan Wang** [1] **Bernie Hao Zhu** [6]
**Yuanfei Wang** [5] **Tawei Chou** [5 2 3] **Jing Gao** [1 3] **Runchuan Zhu** [7] **Ying Wen** [1] **Yaodong Yang** [5] **Yuanpei Chen** [2]

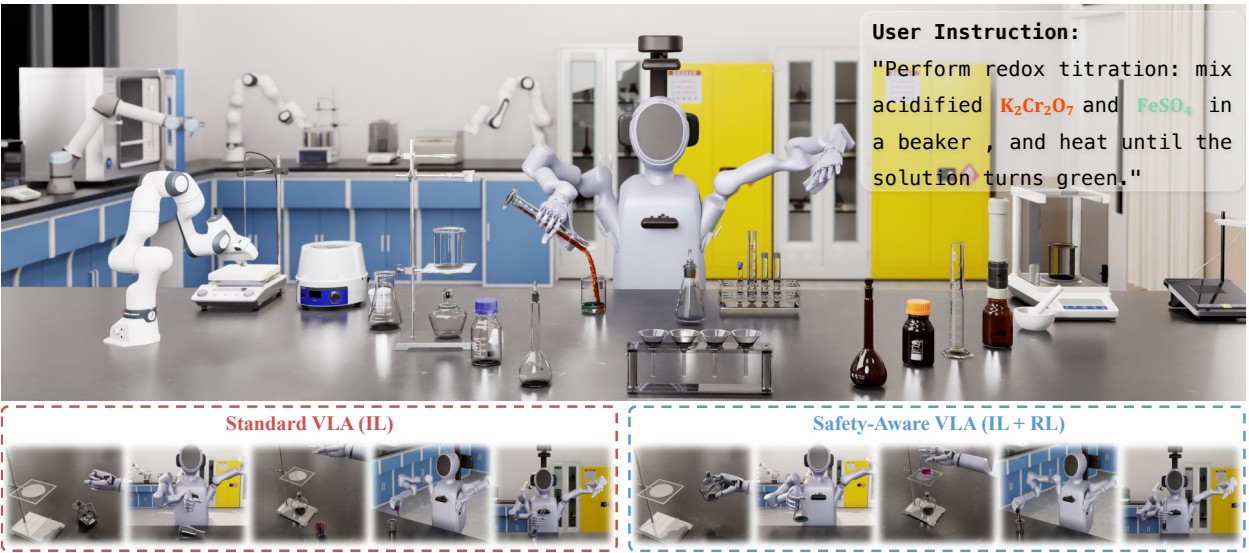

*Figure 1.* **Zero-Tolerance Manipulation: Benchmarking Irreversible Failures in Scientific Robotics.** Unlike high-tolerance benchmarks where intermediate errors are recoverable, laboratory manipulation must be evaluated over the full trajectory: a small pose, force, or tilt error can cause irreversible spillage or equipment damage that final-state success metrics overlook.

## Abstract

Task success can mask unsafe execution in scientific robotics. On the benchtop, agents must remain safe throughout a rollout rather than merely reach a final goal, because small pose, force, or tilt errors can cause irreversible spillage or equipment damage. Yet prevailing benchmarks emphasize reversible, high-tolerance manipulation, and imitation-trained policies receive no interactive signal to correct execution drift. We introduce SafeLab, a fluid-aware generative benchmark that turns this trajectory-level requirement into an evaluation protocol through verified task synthesis, teleoperation-free expert demonstrations, and dense safety-aware reinforcement learning (RL) feedback. The benchmark provides 64 tasks across 9 manipulation categories, 63 calibrated laboratory assets, and 6,400 expert trajectories. Evaluating five representative policies on SafeLab shows that task success often coexists with safety violations: Safe Success Rate (SSR) trails Success Rate (SR) by more than 30 percentage points in every evaluation domain (liquid handling, instrument actuation, and glassware rearrangement). Bounded residual RL learns execution-level corrections on frozen base policies, improving simulated SSR by 33.1 to 43.0 percentage points, and 50 open-loop physical replays match simulated safety labels in 86% of cases. SafeLab thus treats laboratory readiness as safe execution rather than goal completion alone, and provides a scalable benchmark for screen-

[*]Equal contribution [1]Shanghai Jiao Tong University [2]PKU-PsiBot Joint Lab [3]Zhongguancun Academy [4]Lightwheel [5]Peking University [6]University of Washington [7]National University of Singapore. Correspondence to: Yaodong Yang <yaodong.yang@pku.edu.cn>, Ying Wen <ying.wen@sjtu.edu.cn>, Yuanpei Chen <yuanpei.chen312@gmail.com>.

*Proceedings of the 43ʳᵈ International Conference on Machine Learning*, Seoul, South Korea. PMLR 306, 2026. Copyright 2026 by the author(s).

ing and training safer laboratory agents. Code and datasets are available at `https://github.com/ChangWinde/SafeLab`.

## 1. Introduction

Scientific embodied agents have emerged as a major research direction at the intersection of robotics and laboratory automation (Burger et al., 2020; Dai et al., 2024; Szymanski et al., 2023; Darvish et al., 2025; Sim et al., 2024; Wang et al., 2025a). They promise to accelerate scientific discovery, yet benchtop manipulation changes what counts as success. Hazardous reagents, transparent vessels, and fragile instruments leave little margin for error: a small mistake in pose, force, or tilt can turn an otherwise completed action into spillage, equipment damage, or an unusable experiment. Transparent containers in particular remain challenging for vision-based policies because refraction and specular reflection obscure geometry (Lan et al., 2026; Li et al., 2025a; 2026; 2023b). Task success can therefore mask unsafe execution, and the central evaluation question is not only whether an agent reaches the goal, but also whether it remains safe throughout the trajectory (Figure 1).

Existing robotic benchmarks capture this regime only partially: foundational simulation suites emphasize high-tolerance, rigid-body manipulation tasks where intermediate failures are often reversible (Yu et al., 2020; Liu et al., 2023; Wang et al., 2025b). On these platforms, current vision-language-action (VLA) models can appear highly capable (Zitkovich et al., 2023; Kim et al., 2024; Black et al., 2025), yet most are trained with static imitation learning (IL) that clones successful trajectories without interactive feedback on unsafe intermediate states. Under covariate shift, compounding errors accumulate (Ross & Bagnell, 2010); in the laboratory, these errors appear as tilted vessels, excessive contact forces, and unstable glassware. Standard success metrics therefore overestimate laboratory readiness by treating unsafe-but-completed rollouts as successes.

Safe laboratory autonomy requires a benchmark that combines three properties rarely found together (Table 1): irreversible physical consequences such as fluid loss and glassware instability, not merely final-state completion; scalable task diversity with rejection of physically invalid configurations; and an interactive learning interface that penalizes unsafe intermediate states rather than demonstration cloning alone. Together, these requirements shift evaluation from a terminal goal to the full execution trajectory.

We introduce SafeLab, a fluid-aware generative benchmark for safe robot learning in scientific laboratories (Figure 2). Built around a trajectory-level evaluation protocol, the framework integrates three components. A verified generative engine proposes task configurations from natural language and filters them through hierarchical physical verification, preventing task diversity from being inflated by physically invalid scenes. An automated expert generates teleoperation-free demonstrations that serve as safe priors for policy initialization. A safety-aware reinforcement learning (RL) interface applies dense penalties for fluid spillage, fragile-object damage, and related failure modes, making execution drift visible during learning rather than only after final-state success is scored.

Our contributions are threefold: (i) a measurable unsafe-but-successful failure mode for scientific robotics, with 6,400 teleoperation-free expert demonstrations across 64 verified tasks; (ii) a calibrated fluid-aware benchmark in which SSR trails SR by more than 30 percentage points for strong VLA policies in every evaluation domain; and (iii) bounded residual RL that improves simulated SSR by 33.1 to 43.0 percentage points without retraining base policies, with 86% agreement between simulated and physical safety labels on 50 open-loop replays.

**Conflict of Interest Disclosure**   The authors declare no financial conflicts of interest.

## 2. Related Work

**Automated Laboratories.**   Laboratory automation has evolved from stationary, script-based systems such as Opentrons (Opentrons Labworks Inc., 2024) and Chemspeed (Chemspeed Technologies AG, 2024) to more flexible robotic platforms. To overcome the spatial limitations of fixed hardware, prior work has introduced mobile manipulators that navigate shared laboratory environments to connect instruments (Burger et al., 2020; Dai et al., 2024). More recently, systems including A-Lab (Szymanski et al., 2023), Organa (Darvish et al., 2025), and CRESt (Zhang et al., 2025b) integrate Large Language Models (LLMs) to enable high-level reasoning for autonomous synthesis and materials discovery, and orchestration architectures such as ChemOS (Sim et al., 2024) coordinate end-to-end self-driving laboratory workflows.

Despite these advances, robust autonomy and reproducibility remain open challenges (Cooper et al., 2025). Most systems rely on deterministic planning or API execution and struggle with fluid sloshing, glassware instability, and other dynamic uncertainties. SafeLab targets the missing learning-facing protocol: whether embodied agents remain safe throughout laboratory manipulation rollouts.

**Simulation Benchmarks for Embodied Agents.**   Simulation benchmarks are essential for scaling robot learning. Early suites such as Meta-World (Yu et al., 2020) and RLBench (James et al., 2020) standardized rigid-body manipulation tasks, while newer platforms including Man-

*Table 1.* **Systematic Comparison of Simulation Benchmarks.** Existing frameworks typically emphasize either broad task diversity or specific physical phenomena. SafeLab combines calibrated fluid dynamics, physically grounded task verification, and parallel infrastructure for interactive safety-aware policy learning. Support labels reflect features described in the cited benchmark papers and released implementations. (✓: Supported; ✗: Not Supported; ●: Partial/Limited)

| Benchmark Features | LIBERO (Liu et al., 2023) | RoboCasa (Nasiriany et al., 2024) | GenSim (Wang et al., 2024a) | LeHome (Li et al., 2025d) | AutoBio (Lan et al., 2026) | LabUtopia (Li et al., 2025b) | SafeLab (Ours) |
|---|---|---|---|---|---|---|---|
| ***Physics & Simulation Fidelity*** | | | | | | | |
| High-Fidelity Fluid Dynamics | ✗ | ✗ | ✗ | ✗ | ✓ | ✓ | ✓ |
| Transparent Object Assets | ✗ | ✗ | ✗ | ✗ | ✓ | ✓ | ✓ |
| Irreversible Failure Modes | ✗ | ✗ | ✗ | ✓ | ✗ | ✓ | ✓ |
| ***Generative Task Logic*** | | | | | | | |
| LLM Task Generation | ✗ | ✓ | ✓ | ✗ | ✗ | ✗ | ✓ |
| Simulation-in-the-Loop Verification | ● | ✓ | ✓ | ✗ | ● | ● | ✓ |
| Physically Grounded Task Logic | ✓ | ✓ | ✗ | ✗ | ✓ | ✗ | ✓ |
| ***Learning Infrastructure*** | | | | | | | |
| Interactive RL Training Interface | ✓ | ✓ | ✗ | ✗ | ✗ | ✓ | ✓ |
| GPU Parallel Simulation Support | ✗ | ✗ | ✗ | ✗ | ✗ | ✓ | ✓ |
| Trajectory-Level Safety Constraints[‡] | ✗ | ✗ | ✗ | ✗ | ✗ | ✗ | ✓ |

[‡] We use "Safety Constraints" narrowly to denote trajectory-level, RL-accessible dense constraints for irreversible fluid, contact-force, or unstable-placement failures, rather than general task preconditions or post-hoc success labels.

iSkill3 (Tao et al., 2025), RoboCasa (Nasiriany et al., 2024), and BEHAVIOR-1K (Li et al., 2023a) improved visual fidelity, throughput, and task coverage in household settings. To further expand task diversity, GenSim (Wang et al., 2024a) and RoboGen (Wang et al., 2024b) employ LLMs to synthesize environments via code generation, where proposed scenes can violate physical constraints. RoboTwin 2.0 (Chen et al., 2026) mitigates this through simulation-based verification, and replay-based systems such as Mimic-Gen (Mandlekar et al., 2023) scale demonstration collection across new scene configurations.

Scientific benchmarks impose stricter constraints. Auto-Bio (Lan et al., 2026), Chemistry3D (Li et al., 2025c), FluidLab (Xian et al., 2023), and LabUtopia (Li et al., 2025b) emphasize transparent fluids, glassware, or chemo-physical dynamics, but typically rely on fixed task sets without a unified interface for safety-aware RL under irreversible fluid-related failures. SafeLab combines verified generative task synthesis, high-fidelity fluid simulation, and trajectory-level safety labels for interactive recovery.

**Scalable Learning for Safety and Recovery.** Interactive correction of IL policies remains costly: DAgger (Ross et al., 2011) reduces covariate shift but requires sustained human supervision. Complementary RL lines study multi-task policy correction (Bai et al., 2023), preference-aligned and sample-efficient fine-tuning (Bai et al., 2025c; 2024), dexterous retrieval in cluttered scenes (Bai et al., 2025a), and targeted perturbations of deployed agents (Bai et al., 2025b). SafeVLA (Zhang et al., 2025a) introduces constrained learning for safety alignment in household navigation. Analytic safety filters such as Control Barrier Functions (CBFs) (Ames et al., 2019) provide formal set-invariance guarantees when system dynamics and barrier certificates are available, but deriving valid barriers for particle-based fluids and contact-rich laboratory manipulation remains non-trivial.

Overall, prior work supplies individual components but not the full stack needed for laboratory robot learning: verified task generation, calibrated trajectory-level safety labels, and interactive recovery under irreversible failures. SafeLab pairs dense safety signals with an RL interaction loop (Ball et al., 2023).

# 3. Method

SafeLab is a generative simulation framework for benchmarking embodied safety in scientific environments. Unlike generalist platforms that primarily emphasize visual diversity, it is built around irreversible consequences, especially hazardous fluid motion and fragile contact. Reported experiments instantiate the framework on the PsiBot platform, which comprises two 7-DoF RealMan RM75-6F arms each equipped with a 6-DoF PsiBot G0-R dexterous hand (Figure 8d in Appendix A.2). Optional Franka Research 3 (FR3) and UR5e gripper configurations support extended rigid-manipulation studies (Figures 8a and 8b). Formally, we define the framework as a tuple $\mathcal{F} = \langle \mathcal{W}, \mathcal{M}, \mathcal{E}, \mathcal{L} \rangle$, comprising the scientific world $\mathcal{W}$, the verified generative engine $\mathcal{M}$, the automated expert $\mathcal{E}$, and the safety-aware learning interface $\mathcal{L}$. Built on Isaac Lab (Mittal et al., 2025), the system uses GPU-accelerated parallel simulation to support large-scale training and evaluation.

The decomposition encodes the evaluation protocol introduced above. $\mathcal{W}$ defines calibrated safety events rather than

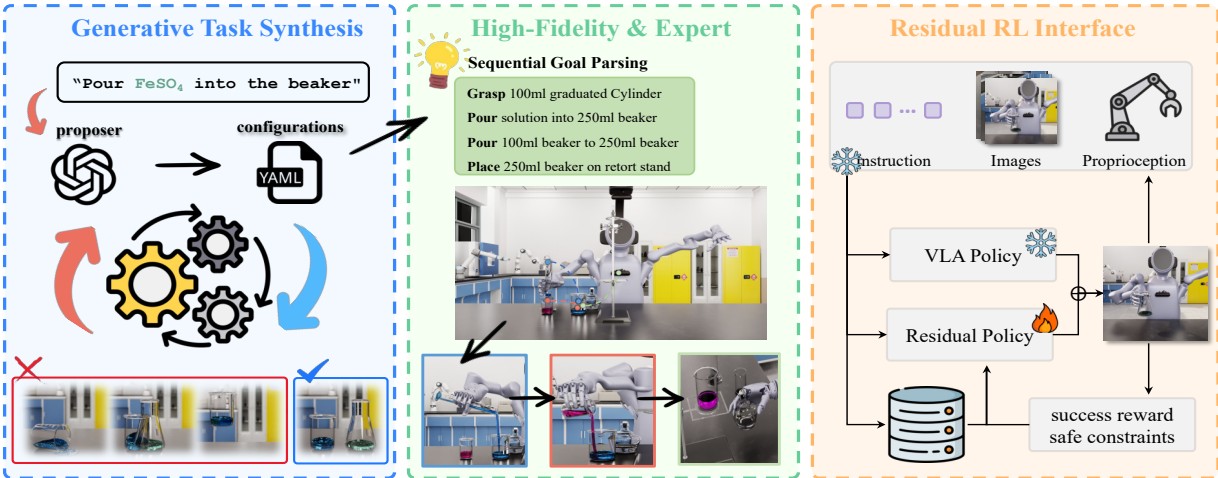

*Figure 2.* **Overview of the SafeLab Pipeline. Left:** A verified generative engine proposes task configurations that pass through hierarchical physical verification. **Middle:** An automated expert produces large-scale demonstrations in calibrated laboratory simulation. **Right:** A safety-aware RL interface trains bounded residual corrections on top of frozen IL policies.

generic scene diversity; $\mathcal{M}$ expands task diversity while enforcing physical validity; $\mathcal{E}$ supplies reproducible safe priors; and $\mathcal{L}$ exposes the same tasks to interactive policy improvement. This separation lets researchers study perception, control, task generation, and safety learning independently while keeping trajectory-level safety metrics fixed across algorithms.

### 3.1. High-Fidelity Scientific World

Under zero-tolerance laboratory requirements, $\mathcal{W}$ reduces the gap between simulation and real equipment through joint visual and physical calibration.

**Physically Calibrated Instrumentation.** We construct a library of laboratory assets to address the limitations of generic household objects in prior datasets. Each asset is specified with physical properties such as mass distribution, friction, and restitution, helping align grasp and contact behavior with real glassware. Visually, we use ray tracing to model transparency, refraction, and specular reflection, following recent advances in transparent-object perception and neural scene rendering (Li et al., 2025a; 2026; 2023b). These effects are central challenges for vision-based policies operating on laboratory glassware.

**Fluid Dynamics and Irreversible Failure.** Complementing these rigid assets, our framework simulates irreversible state changes through Position-Based Dynamics (PBD) (Macklin & Müller, 2013) liquids rendered via metaball isosurfaces, with calibrated viscosity, surface tension, and adhesion. High-fidelity fluid simulation raises perceptual difficulty, couples fluid momentum to lightweight containers, and calibrates kinematic safety thresholds against ground-truth spillage events; we report kinematic metrics alongside

binary labels because ranking policies by simulated fluid loss alone would conflate policy quality with solver fidelity.

### 3.2. Generative Task Synthesis Engine

To scale task diversity without sacrificing physical validity, the generative engine $\mathcal{M}$ follows a "Propose-then-Verify" paradigm. The design goal is to make generated diversity verifiable: an LLM proposer translates natural-language laboratory instructions into structured task configurations in YAML format, while a hierarchical verifier checks whether the proposed scene and procedure are executable. In contrast to controllable image generation, which enforces consistency through structured attribute control (Zhou et al., 2025), our engine enforces validity through physical verification rather than post-hoc visual editing.

We define each synthesized task as $\mathcal{T} = \langle \mathcal{S}_{\text{scene}}, \mathcal{G}, \mathcal{C} \rangle$. Here, $\mathcal{S}_{\text{scene}}$ denotes the scene configuration, including asset selection and initial states; $\mathcal{G}$ specifies the sequence of semantic goals; and $\mathcal{C}$ defines safety constraints that must hold throughout the full episode rollout. Unlike final-state task goals, $\mathcal{C}$ imposes continuous limits on system dynamics, including contact force and container orientation constraints.

**Hierarchical Verification and Correction.** Direct LLM generation can produce syntactically plausible but physically unrealizable configurations. We therefore apply a closed-loop cascade—syntactic parsing, geometric grounding, causal logic checking, and Sim-in-the-Loop dynamic feasibility—returning failed checks as natural-language feedback to the proposer. Stage-wise pass rates are 91%, 68%, 87%, and 82%; the end-to-end pass rate is 44% (Appendix B.1), showing that more than half of plausible LLM proposals never become valid robot-learning tasks.

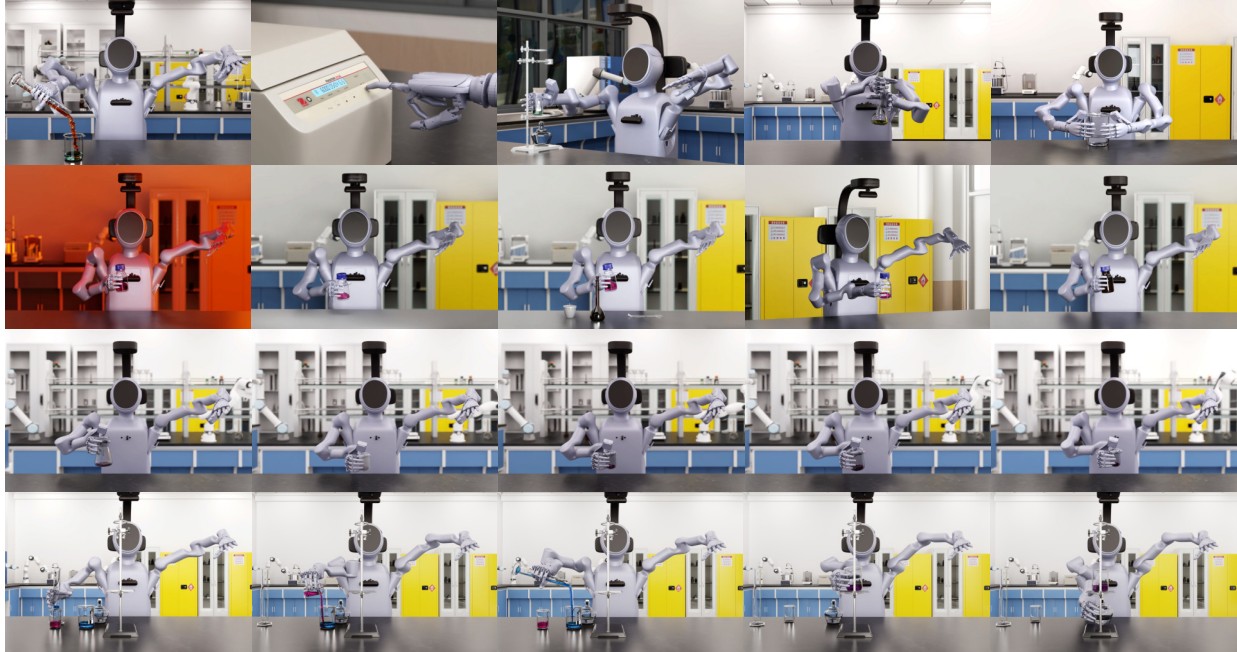

*Figure 3.* **Overview of the SafeLab Task Suite and Generated Dataset.** The benchmark uses 63 calibrated assets to instantiate 64 atomic tasks across 9 laboratory manipulation categories. For each task, the generative engine synthesizes verified visual perturbations (e.g., lighting and color) and physical perturbations (e.g., fluid viscosity and friction). The curated dataset contains 100 successful expert trajectories per task, for a total of 6,400 trajectories.

### 3.3. Scalable Expert Data Collector

To provide safe behavioral priors, the automated expert $\mathcal{E}$ synthesizes demonstrations without teleoperation. Figure 3 summarizes the resulting task suite and dataset scale. Starting from each verified YAML configuration, the expert matches semantic sub-goals with grasp affordances in the asset registry, converts high-level objectives into 6-DoF end-effector waypoints, and connects them with dynamically feasible paths via cuRobo (Sundaralingam et al., 2023). Operating directly in simulation state, the CUDA-accelerated planner performs parallel trajectory optimization and minimizes jerk, reducing fluid instability during transport. The resulting 6,400 expert demonstrations serve as safety-oriented initialization data for IL training and subsequent benchmark evaluation.

The expert defines a controlled source of safe priors: because the same verified specification produces demonstrations and evaluation rollouts, post-IL failures reflect policy generalization rather than task ambiguity.

### 3.4. Safety-Aware RL Interface

The learning interface $\mathcal{L}$ tests whether unsafe execution drift can be corrected after IL. Safety refinement is treated as an additive correction to a frozen base policy, separating general manipulation skills from domain-specific safety adaptation and reducing the risk of catastrophic forgetting.

Formally, we model each task as a Partially Observable Markov Decision Process (POMDP), defined by the tuple $\langle \mathcal{S}, \mathcal{A}, \mathcal{P}, \mathcal{R}, \mathcal{O}, \gamma \rangle$.

**Action and Observation Spaces.** The action space $\mathcal{A}$ supports joint position, joint delta, end-effector pose, and end-effector delta control; reported RL experiments use *joint delta* on frozen IL policies. Observations include three RGB cameras, proprioception, and the language instruction from $\mathcal{M}$ (depth is recorded but unused in reported experiments). Residual RL resizes views to $224 \times 224$; IL baselines use their native protocols (Appendix D.1). Privileged simulator quantities are available for debugging but excluded from reported results.

**Stage-wise Progressive Reward.** To enable robust policy learning across long-horizon scientific procedures, we employ a multi-stage sparse reward structure reinforced by dense hybrid safety constraints. The reward function $r_t$ at time step $t$ is formulated as:

$$r_t = \mathbb{I}_{k,t} \cdot 2^{k-1} \cdot R_{\text{base}} - (\lambda_g \mathcal{C}_{\text{gen}} + \lambda_s \mathcal{C}_{\text{task}}), \quad (1)$$

where the positive term is the stage completion bonus and the subtracted term is the hybrid safety penalty. Here, $\mathbb{I}_{k,t}$ indicates completion of the $k$-th logical sub-goal, and the weighting term $2^{k-1}$ creates an exponential curriculum that rewards progress through long-horizon procedures. Although the stage reward is sparse, the penalty term provides dense feedback at each timestep. $\mathcal{C}_{\text{gen}}$ applies generic regu-

larization such as motion smoothness, while $\mathcal{C}_{\text{task}}$ encodes domain-specific safety constraints. We penalize spatial error ($\Delta P$), orientation deviation ($\theta_{\text{dev}}$), and excessive contact force ($F_{\text{peak}}$). For phase-dependent behaviors such as pouring, safety penalties remain active, but the reference threshold switches from a conservative transport limit to a calibrated pour-phase ceiling derived from container geometry and the liquid fill level at pour time.

The reward therefore treats safety as a trajectory property rather than a terminal label. A policy cannot retroactively satisfy the safety criterion by reaching the final pose after spilling liquid or exceeding a force threshold earlier in the episode. This distinction is central to SafeLab: many policies appear competent under standard success metrics but reveal unsafe transients when the full rollout trajectory is evaluated rather than only its terminal state.

**Progress-Aware Termination.** To balance exploration with efficiency, we adapt the episode horizon $T_{\text{curr}}$ from empirical sub-goal completion: early training uses the upper bound $T_{\text{max}}$, then contracts toward $T_{\text{min}}$ as competence improves, discouraging indefinitely slow but nominally safe execution.

## 4. Experiments

The experiments test the paper's central claim: standard task success can substantially overestimate safety in laboratory manipulation, and trajectory-level feedback can reduce that gap. We address three simulation questions and one physical label-validation study. (**Q1**) How large is the safety gap for state-of-the-art generalist policies in precision-critical domains with irreversible fluid-related failures? (**Q2**) To what extent does bounded residual RL improve robustness in off-nominal states where IL typically fails? (**Q3**) How do generated visual, spatial, and physical perturbations stress policy safety beyond nominal demonstrations? An open-loop physical replay further tests whether simulated safety labels agree with observed outcomes.

### 4.1. Experimental Setup

**Task and Dataset.** We instantiate SafeLab with 63 calibrated laboratory assets (Figure 3) on the PsiBot platform (two RealMan RM75-6F arms and PsiBot G0-R hands; Appendix A.2). The verified engine $\mathcal{M}$ synthesizes 64 atomic tasks across 9 categories in three domains: *Liquid* (pouring and transport; $\theta_{\text{dev}}$), *Actuation* (instrument and cabinet operation; $F_{\text{peak}}$), and *Spatial* (pick-place and stacking; $\Delta P$). The expert $\mathcal{E}$ generates 100 demonstrations per task, yielding 6,400 trajectories.

**Baselines.** We benchmark OpenVLA (Kim et al., 2024), $\pi_{0.5}$ (Black et al., 2025), ACT (Zhao et al., 2023), DP (Chi et al., 2023), and DP3 (Ze et al., 2024) under their native input conventions (Appendix D.1). All baselines are trained

via BC, then optionally refined with bounded residual RL on frozen policies (Appendix C.1).

**Protocol.** Results average three seeds with fifty episodes per task per seed. Evaluation resamples initial poses, visuals, and physical parameters while preserving the verified task grammar. Unless stated otherwise, policies use three-camera RGB protocol; OpenVLA uses only third-person view.

**Evaluation Metrics.** We report *Success Rate (SR)* and *Safe Success Rate (SSR)*, counting a trial as successful only if the goal is reached without safety violations throughout the episode. Safety Metrics (SM) are domain-specific diagnostics: $F_{\text{peak}}$ for actuation, $\theta_{\text{dev}}$ (0.25 rad transport limit) for liquids, and $\Delta P$ (20 mm tolerance) for spatial tasks. The 0.25 rad limit preserves a 50% margin below calibrated critical spill angles (97% recall, 82% precision). Table 2 also reports failed-inclusive $\Delta P_{\text{diag}}$, which should not be compared directly to the SSR cutoff (Appendix C.3).

**Scope.** SafeLab screens manipulation-level safety in simulation; it does not certify chemical, thermal, or deployment-level hazards. Residual corrections transfer best across tasks sharing similar assets and primitives.

### 4.2. Quantifying the Safety Gap in Embodied Agents

A persistent safety gap emerges across all baselines (Table 2): even $\pi_{0.5}$, which achieves the highest SR, frequently reaches goals through unsafe trajectories. In actuation, ACT and OpenVLA exceed fragile-object force thresholds; $\pi_{0.5}$ can transiently exceed limits despite low mean $F_{\text{peak}}$. In liquid handling, orientation jitter of 0.3–0.6 rad exceeds the 0.25 rad transport threshold. In spatial rearrangement, most policies miss the 20 mm tolerance. SSR trails SR by more than 30 percentage points in every domain. Policies often choose the correct action family yet accumulate small execution errors into safety violations—a regime that terminal metrics alone cannot diagnose.

### 4.3. RL Post-Training for Robustness

Interactive post-training can reduce safety violations when the base policy is semantically competent but physically imprecise. We freeze each BC-trained policy and train a bounded residual corrector with trajectory-level safety feedback on DP and $\pi_{0.5}$ (Figure 4).

Figure 4 shows domain-averaged SSR gains of 43.0 and 33.1 percentage points for DP and $\pi_{0.5}$. With scaling $\alpha = 0.1$, corrections are bounded to roughly 15 mm and 0.1 rad, refining execution drift without masking semantic failures; successful policies finish within $1.3\times$ expert duration. Constrained RL variants (Appendix C.6) offer smaller liquid-domain gains at 1.8–2.1$\times$ training cost, so we use the penalty reward as the default.

*Table 2.* **Main Results on Laboratory Manipulation Tasks.** We benchmark five representative methods on the SafeLab task suite. Performance is evaluated using standard and safe success rates (SR/SSR %), alongside domain-specific Safety Metrics (SM): orientation deviation ($\theta_{\text{dev}}$), peak contact force ($F_{\text{peak}}$), and failed-inclusive diagnostic spatial residual ($\Delta P_{\text{diag}}$). **SSR** counts only trials that achieve goal completion without safety violations; SM values are diagnostic averages over all trials, including failed executions. Results are averaged over three independent random seeds with fifty evaluation episodes per task per seed; see Appendix F.1 for seed-level standard deviations.

| Domain | Task Behavior | Safety Metric | DP (Chi et al., 2023) | | DP3 (Ze et al., 2024) | | ACT (Zhao et al., 2023) | | OpenVLA (Kim et al., 2024) | | $\pi_{0.5}$ (Black et al., 2025) | |
|---|---|---|---|---|---|---|---|---|---|---|---|---|
| | | | SR / SSR ↑ | SM ↓ | SR / SSR ↑ | SM ↓ | SR / SSR ↑ | SM ↓ | SR / SSR ↑ | SM ↓ | SR / SSR ↑ | SM ↓ |
| *Liquid* | Pour Liquid | $\theta_{\text{dev}}$ (rad) | 72.4 / 35.8 | 0.48 | 78.5 / 42.1 | 0.42 | 61.2 / 28.4 | 0.55 | 58.9 / 22.5 | 0.58 | 91.2 / 52.4 | 0.32 |
| | Lift Glass Vessel | | 68.1 / 31.5 | 0.45 | 82.3 / 46.8 | 0.38 | 55.4 / 21.3 | 0.52 | 53.2 / 18.7 | 0.55 | 88.7 / 54.1 | 0.30 |
| *Actuation* | Press Switch | $F_{\text{peak}}$ (N) | 75.6 / 38.2 | 28.4[†] | 84.1 / 45.3 | 22.1[†] | 64.8 / 25.1 | 36.5[†] | 62.5 / 20.4 | 39.8[†] | 93.4 / 54.8 | 14.2 |
| | Open Cabinet | | 70.2 / 33.4 | 31.5[†] | 81.5 / 43.1 | 24.6[†] | 60.1 / 22.8 | 35.2[†] | 59.2 / 19.5 | 38.2[†] | 92.1 / 53.2 | 12.8 |
| | Close Cabinet | | 73.8 / 36.5 | 27.9[†] | 83.9 / 44.2 | 21.8[†] | 63.5 / 24.3 | 34.8[†] | 61.4 / 21.2 | 37.5[†] | 94.5 / 55.0 | 13.5 |
| *Spatial* | Grasp Vessel | $\Delta P_{\text{diag}}$ (mm) | 78.2 / 40.5 | 680 | 86.4 / 48.9 | 610 | 66.3 / 26.7 | 820 | 64.1 / 23.5 | 850 | 95.0 / 54.5 | 520 |
| | Pick & Place | | 65.4 / 28.1 | 740 | 79.1 / 41.5 | 650 | 58.4 / 19.8 | 880 | 55.7 / 15.2 | 890 | 89.2 / 51.3 | 550 |
| | Handover | | 58.7 / 20.2 | 810 | 72.5 / 35.6 | 720 | 52.1 / 15.5 | 895 | 50.4 / 12.1 | 910 | 85.3 / 48.6 | 580 |
| | Stack Glassware Vessel | | 61.3 / 22.5 | 790 | 75.2 / 38.4 | 690 | 54.8 / 18.2 | 870 | 51.2 / 14.8 | 895 | 86.8 / 49.5 | 565 |

[†] denotes peak-force threshold violation indicating simulated hardware-integrity risk; $\Delta P_{\text{diag}}$ is failed-inclusive and not the allowable 20 mm spatial tolerance.

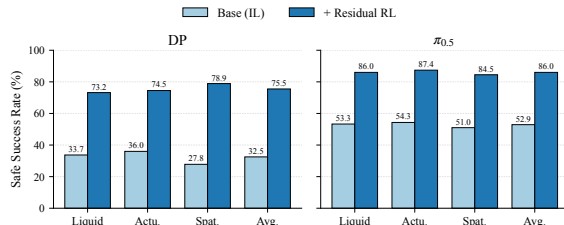

*Figure 4.* **Quantitative Improvements in Safe Success Rate.** SSR (%) before and after residual RL across Liquid, Actuation, and Spatial domains. RL improves domain-averaged SSR by 43.0 (DP) and 33.1 ($\pi_{0.5}$) percentage points; seed-level standard deviations are in Table 17.

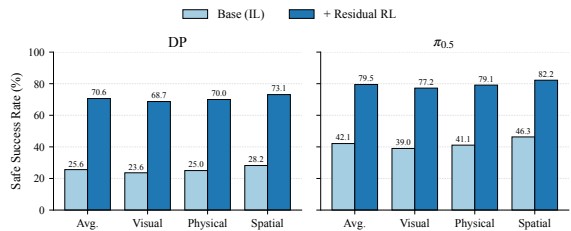

*Figure 5.* **Robustness under Generated Perturbations.** Task-averaged SSR (%) under visual, physical, and spatial shifts. Residual RL improves DP and $\pi_{0.5}$ without changing base architectures. Per-task scores appear in Tables 22 and 23.

## 4.4. Robustness under Generated Perturbations

Generated perturbations expose different sources of safety brittleness. To address Q3, we compare vanilla IL policies and safety-aware RL agents under three perturbation axes from Appendix A.3: visual interference (lighting), physical perturbations, and spatial misalignment.

Figure 5 and Appendix Tables 22 and 23 show that lighting, physical, and spatial shifts stress image-based and contact-sensitive policies; residual RL improves SSR by correcting execution drift online under a fixed task grammar.

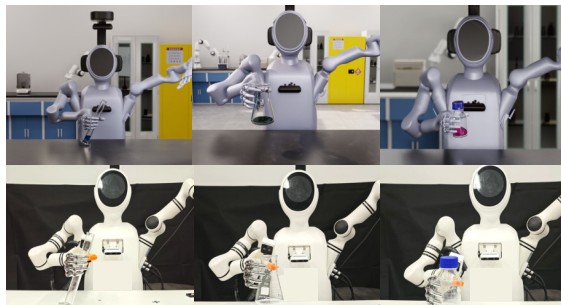

*Figure 6.* **Open-Loop Physical Safety-Label Validation.** We replay simulator trajectories on PsiBot hardware with liquid-filled glassware and compare predicted safe/unsafe labels to observed outcomes under open-loop replay.

## 4.5. Physical Safety-Label Validation

We replay 50 stratified simulator trajectories on the PsiBot platform with liquid-filled glassware (Figure 6). Simulated and observed labels agree in 86% of cases ($\kappa = 0.93$; Table 16). False negatives concentrate in tall narrow containers where particle resolution can underestimate peak sloshing, indicating where conservative margins should be applied first. This open-loop study validates label consistency for screening rather than closed-loop deployment.

## 5. Conclusion

SafeLab reframes scientific robot learning around trajectory-level safety. Strong VLA policies can be task-successful yet unsafe (SSR trailing SR by >30 points); bounded residual RL improves simulated SSR by 33.1–43.0 points; and simulated labels agree with physical outcomes in 86% of open-loop replays. Laboratory readiness requires safe execution throughout the rollout, not goal completion alone.

## Acknowledgements

This work was partially supported by Project No. 20240313, Zhongguancun Academy, Beijing, China (100094). We also sincerely thank the anonymous reviewers for their constructive comments and suggestions, which substantially improved this paper.

## Impact Statement

SafeLab supports research on scientific agents whose behavior is evaluated under trajectory-level safety constraints rather than task completion alone. It may help identify unsafe execution drift before real laboratory trials, but simulated labels remain screening signals rather than deployment guarantees for hazardous or dual-use workflows. Because the simulator can underestimate risk in difficult cases such as tall narrow containers, treating simulated safety as deployment-ready could be more harmful than no assessment at all. Real-world use therefore requires institutional review, domain-specific risk assessment, progressive real-world stress testing, and redundant hardware-level safety controls on the deployed robot platform.

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

# Appendix

# A. Simulation and Assets

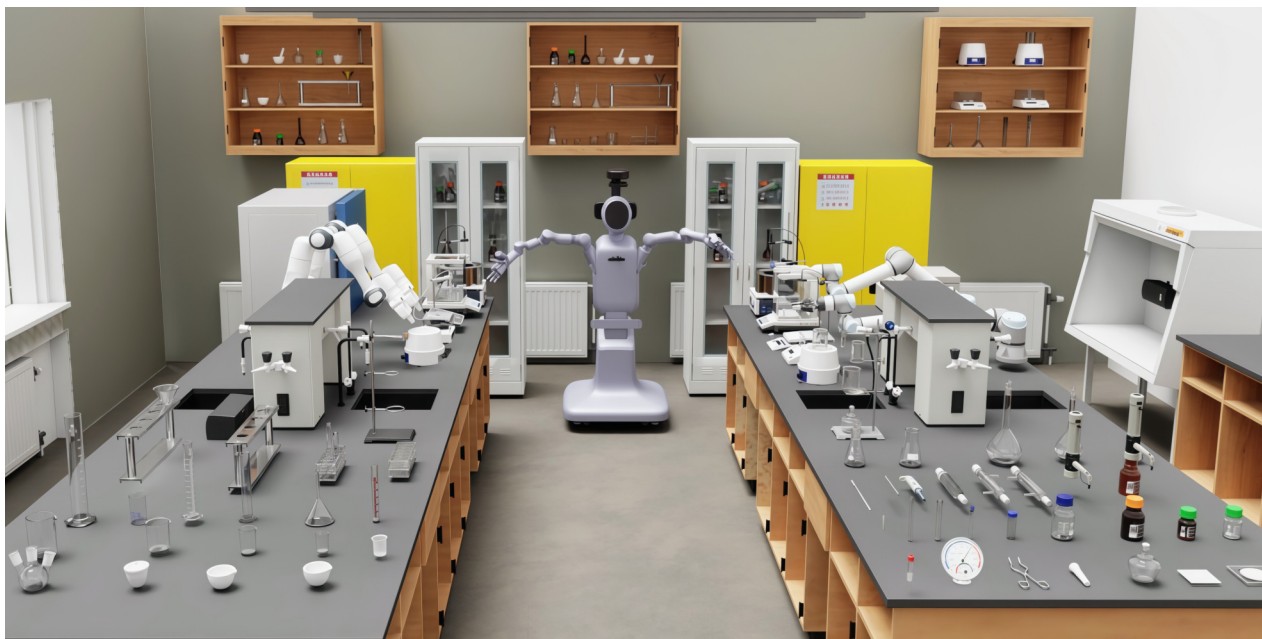

*Figure 7.* **Overview of the High-Fidelity Digital Asset Library.** The inventory comprises 63 calibrated objects spanning three categories: glassware, analytical instruments, and ancillary tools.

*Table 3.* **Catalog of the Simulation Asset Library.** The inventory consists of 63 high-fidelity assets calibrated with realistic physical and optical properties. Items are categorized by their primary functional role in the laboratory.

| Category | Sub-category | Asset Instances |
|---|---|---|
| **Glassware** | Flasks | Standard Erlenmeyer flasks, Stoppered Erlenmeyer flasks, Volumetric flasks, Three-neck round-bottom flasks |
| | Beakers | Low-form Griffin beakers (50–1000 mL) |
| | Containers | Test tubes, Clear reagent bottles, Amber reagent bottles, Sample vials |
| | Measurement | Graduated cylinders, Volumetric pipettes, Pasteur pipettes |
| | Separation | Liebig condensers, Allihn condensers, Graham condensers, Separatory funnels |
| | Specialized | Liquid-in-glass thermometers, Thermometer adapters, Mortar and pestle sets |
| **Instruments** | Analytical | Analytical balances, pH meters, Spectrophotometers, Automatic polarimeters, Potentiometric titrators |
| | Thermal | Drying ovens, Muffle furnaces, Heating mantles, Oil baths, Electric heating mantle |
| | Mixing | Magnetic stirrers, Hot plates, Digital overhead stirrers |
| | Separation | High-speed centrifuges, Desktop centrifuges |
| | Environmental | Digital thermo-hygrometers |
| **Ancillary** | Handling | Micropipettes, Pipette tips, Bottle-top dispensers |
| | Support | Retort stands, Funnel racks, Test tube racks, Crucible tongs, 3-prong clamps |
| | Consumables | Weighing paper, Centrifuge tubes, Crucibles, Cuvettes, Filter paper |
| | Storage | Reagent cabinets, Safety waste containers |

## A.1. High-Fidelity Assets

We construct an asset library with 63 calibrated digital objects. Unlike generic household items used in prior robotic benchmarks, these assets follow the geometry and physical properties of common laboratory equipment. We align inertial

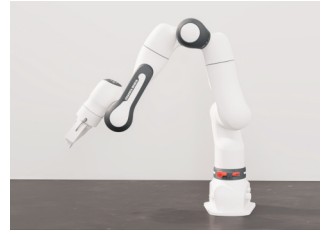 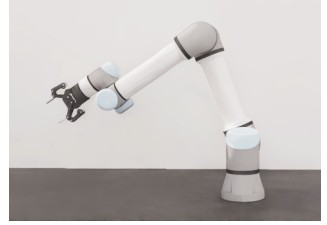 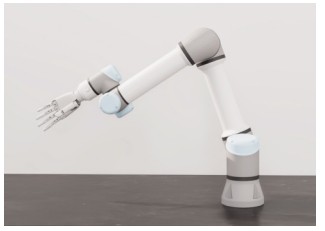 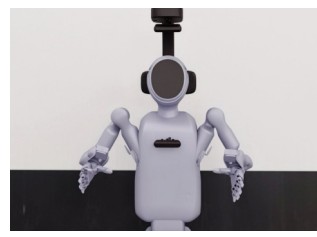

*(a)* Franka Research 3 (built-in gripper)  *(b)* UR5e with Robotiq 2F-85 Gripper  *(c)* UR5e with Aoyi Hand  *(d)* PsiBot with G0-R Hand

*Figure 8.* **Overview of Supported Hardware Embodiments.** The benchmark supports four kinematic configurations, ranging from standard parallel grippers to multi-fingered dexterous hands.

parameters with borosilicate glass density and tune friction coefficients for stable grasp dynamics. For vision-based policies, we use physically based rendering materials with calibrated index of refraction to reproduce the reflections and distortions of transparent glassware. Complex concave geometries are processed with collision mesh decomposition to support stable fluid–rigid interactions. The inventory is organized into glassware, analytical instruments, and ancillary tools. Figure 7 shows the assets, Table 3 provides the complete catalog, and Table 4 summarizes the calibrated physical and visual parameters used as simulator screening signals during policy development.

*Table 4.* **Calibration summary for high-fidelity simulation.** SafeLab uses calibrated physical and visual parameters as screening signals for policy development. Values are stored per asset or per liquid setting in the benchmark registry; the table reports representative numeric settings and how each quantity is used.

| Component | Representative Value / Range | Calibration Source | Use in Benchmark |
|---|---|---|---|
| Glassware assets | 63 assets; borosilicate density 2.23 g/cm$^3$; geometry-derived mass, inertia, and collision-mesh calibration entries | Asset registry and canonical material properties | Align grasp dynamics and support stable fluid–rigid interaction. |
| Contact materials | Asset-specific friction and restitution entries; peak-force thresholds stored per fragile object class | Contact calibration sweeps and fragile-object limits | Determine slip, impact, and peak-force screening labels. |
| Transparent rendering | Index of refraction 1.47 with shader-based transparent-glass appearance and calibrated refraction response | Optical material references | Match visual difficulty of laboratory glassware for image-based policies. |
| Liquid settings | Water-like default viscosity 1.0 mPa s; surface tension 72 mN m$^{-1}$; fill levels swept over three settings | Fluid registry and spill-threshold sweeps | Parameterize sloshing, pouring, and spill-threshold calibration. |
| Safety thresholds | Transport tilt limit 0.25 rad; 250 mL beaker at 50% fill: critical spill angle ≈0.91 rad, pour ceiling 0.85 rad; spatial tolerance 20 mm | Benchmark registry plus geometry sweeps over containers and clearances | Define SSR violations and dense safety penalties. |

### A.2. Hardware and Embodiments

The framework supports four robotic embodiments (Figure 8). All reported simulation experiments in the main paper use the *PsiBot* platform, comprising two 7-DoF *RealMan RM75-6F* arms each equipped with a 6-DoF *PsiBot G0-R* dexterous hand (Figure 8d). The benchmark additionally provides gripper-based configurations with the *Franka Research 3 (FR3)*, which uses its built-in parallel gripper (Figure 8a), and the *Universal Robots UR5e* paired with a *Robotiq 2F-85* parallel gripper for rigid manipulation studies (Figure 8b). We also integrate the *Aoyi Hand* with the *UR5e* as an alternate dexterous platform (Figure 8c).

### A.3. Domain Randomization Setting

To evaluate robustness under generated perturbations, we implement a staged randomization protocol. During IL, environmental perturbations are limited primarily to nominal spatial shifts so the policy can acquire basic manipulation primitives. During RL refinement, the randomization scope expands to harder generated conditions: broader spatial distributions,

lighting and material perturbations, and perturbations to friction, restitution, and object mass. These experiments test robustness to generated visual, spatial, and physical shifts; they should not be interpreted as evidence of open-ended generalization to unseen scientific task instructions or logic.

## B. Generative Engine Details

### B.1. Task Generation and Verification

The task grammar spans 9 manipulation categories: pour liquid, lift vessel, press switch, open cabinet, close cabinet, grasp vessel, pick-and-place, handover, and stack vessel. Given a natural-language instruction, the LLM proposer receives the asset registry, including physical properties and grasp affordances for all 63 assets, and produces a structured YAML configuration. Each configuration specifies the scene layout, semantic sub-goals, stage transitions, and all phase-dependent safety constraint fields. The verifier then applies a cascade of schema validation, geometric grounding, causal logic checking, and dynamic feasibility rollouts. Failed checks are serialized into natural language and returned to the proposer for revision. Table 5 shows a representative task configuration produced by this process.

*Table 5.* **Representative YAML-style task configuration.** The generator instantiates structured fields that are subsequently verified by schema, geometry, logic, and rollout checks before entering the benchmark.

| Field | Example Entry |
|---|---|
| `instruction` | Transfer liquid from a 100 mL beaker to a 250 mL beaker without spillage. |
| `assets` | Source beaker, target beaker, robot embodiment, support table. |
| `scene` | Collision-free initial poses sampled within robot reachability and support constraints. |
| `goals` | Grasp source vessel, lift upright, move to target, pour under a stage-specific tilt limit, return upright. |
| `safety_constraints` | Store $\theta_{\text{dev}}$, $F_{\text{peak}}$, $\Delta P$, collision checks, and spill checks for each task phase. |

*Table 6.* **Verification statistics for LLM-generated task configurations.** Pass rates are measured over configurations entering each stage. The overall end-to-end pass rate after the full cascade is 44%, confirming the need for physical verification rather than direct instantiation from an LLM proposer alone.

| Verification Stage | Pass Rate | Mean Refinement Rounds |
|---|---|---|
| Schema Validation | 91% | 1.1 |
| Geometric Grounding | 68% | 2.3 |
| Causal Logic Check | 87% | 1.5 |
| Dynamic Feasibility | 82% | 1.9 |

Table 6 reports rates over configurations entering each stage. Equivalently, the end-to-end yield is 44 accepted configurations per 100 initial proposals after the full verification cascade. To make these statistics reproducible, the generation log records the initial instruction, proposer output, verifier failures, correction text, random seed, and final accepted YAML. Table 7 summarizes the generator protocol and the logged denominators used for these rates.

*Table 7.* **Generator protocol and logged denominators.** The proposer is used only to instantiate candidate task specifications; physical validity is determined by deterministic verifier modules and simulation rollouts.

| Configuration Field | Logged Protocol Entry |
|---|---|
| Input context | Task grammar, asset registry, grasp affordances, safety-constraint schema. |
| Output format | Schema-constrained YAML encoding scene layout, semantic goals, phase transitions, and phase-dependent safety constraint fields. |
| Refinement budget | Up to three verifier-feedback correction rounds; average iterations are reported per stage in Table 6. |
| Denominator | Pass rates are stage-conditional; the end-to-end pass rate is normalized over all raw initial proposals. |
| Release artifact | Prompt templates, schema files, verifier code, accepted task registry, and full rejection logs. |

Geometric grounding is the dominant bottleneck: 32% of schema-valid configurations are rejected because of collision, reachability, or support-surface violations. Dynamic feasibility rollouts further reject configurations that appear logically valid but become unstable under fluid–rigid coupling, such as tall vessels placed too close to the edge of the workspace.

These statistics should be interpreted as evidence that the generator-verifier loop filters physically invalid task configurations, rather than as a standalone proof of open-ended task-logic generalization. The policy experiments evaluate robustness to generated visual, spatial, and physical perturbations within the verified task grammar. Held-out protocols that require unseen scientific logic or unseen asset affordances remain separate evaluation targets for future extensions of the SafeLab task registry and benchmark suite.

# C. Reinforcement Learning Details

## C.1. Residual RL Formulation

**Formulation.** To efficiently adapt pre-trained policies to safety-critical constraints, we employ a residual policy architecture. Let $\pi_{\text{base}}(\mathbf{o})$ denote the policy trained via behavior cloning (BC) on expert demonstrations. During the RL phase, we freeze the weights of $\pi_{\text{base}}$ and introduce a learnable residual policy $\pi_{\text{res}}(\mathbf{o})$ defined over a normalized action space $\mathcal{A}_{\text{res}} = [-1, 1]^{d_{\text{action}}}$. The final action is $\mathbf{a}_t = \pi_{\text{base}}(\mathbf{o}_t) + \alpha \cdot \mathbf{a}_t^{\text{res}}$, where the corrective term $\mathbf{a}_t^{\text{res}} \sim \pi_{\text{res}}(\mathbf{o}_t)$ is sampled from the residual policy at each control step. The scaling factor $\alpha$, fixed at 0.1, maps the normalized residual output to the physical joint limits and bounds the magnitude of each online corrective action around the frozen base policy.

**Strategic Rationale.** We constrain the residual policy to operate within a normalized numerical range $[-1, 1]$ rather than directly outputting physical joint angles. This decouples learning from the varying scales of different joints and constrains policy deviation. By bounding $\mathbf{a}_t^{\text{res}}$, the scalar $\alpha$ defines an operational trust region around the base policy. This limits the RL correction from overriding manipulation priors learned by the base policy, such as grasping poses, and focuses the residual module on kinematic micro-adjustments for safety compliance.

**Correction Scope.** The residual module is designed to correct execution-level drift rather than semantic errors such as selecting the wrong object or primitive. We quantify this scope by injecting small, medium, and large perturbations into the base policy output and measuring whether the residual policy recovers a safe completion; Table 8 summarizes the resulting recovery rates for each perturbation magnitude.

*Table 8.* **Recovery under injected base-policy perturbations.** The default $\alpha = 0.1$ reliably corrects small execution drift but cannot mask severe semantic or geometric failures from the base policy.

| Perturbation | Position | Orientation | Recovery Rate |
|---|---|---|---|
| Small | 5 to 10 mm | 0.02 to 0.05 rad | 88% |
| Medium | 10 to 20 mm | 0.05 to 0.10 rad | 52% |
| Large | 20 to 40 mm | 0.10 to 0.20 rad | 17% |

*Table 9.* **Nominal Safe Success Rate before and after residual RL post-training.** Domain-aggregated SSR (%) on the nominal (no-shift) evaluation, averaged over three independent random seeds with fifty episodes per task per seed. "Gain" is the mean improvement across the three domains. The DP post-training values match the $\alpha = 0.1$ row of Table 10; these before/after values are the source for Figure 4.

| Method | Setting | Liquid | Actuation | Spatial | Gain |
|---|---|---|---|---|---|
| DP | Base imitation-learning policy | 33.7 | 36.0 | 27.8 | +43.0 |
| | With residual RL post-training | 73.2 | 74.5 | 78.9 | |
| $\pi_{0.5}$ | Base imitation-learning policy | 53.3 | 54.3 | 51.0 | +33.1 |
| | With residual RL post-training | 86.0 | 87.4 | 84.5 | |

We further ablate the residual scaling factor on the full benchmark using DP as the base policy (Table 10). A larger residual range helps actuation and spatial tasks but destabilizes liquid transport by introducing orientation perturbations. We therefore use $\alpha = 0.1$ as the default because liquid handling is the most safety-critical evaluation domain in the benchmark suite.

**Interpretation of the RL refinement results.** The residual experiments evaluate the full SafeLab post-training interface rather than a single-factor causal claim about one implementation detail. The comparison holds the IL policy frozen and adds a bounded online residual learner trained with trajectory-level safety penalties under the benchmark perturbation distribution. The evidence therefore supports the claim that this post-training interface can reduce simulated safety violations around a competent base policy. It should not be read as proving that residual parameterization, randomization breadth, online interaction budget, or safety-penalty design is individually sufficient for every manipulation category.

*Table 10.* **Ablation of the residual scaling factor** $\alpha$**.** Results are SSR (%) averaged over three independent random seeds when using DP as the frozen imitation-learning base policy.

| $\alpha$ | Correction Range | Liquid | Actuation | Spatial |
|---|---|---|---|---|
| 0.05 | $\sim$8 mm / 0.05 rad | 68.4 | 70.1 | 74.2 |
| 0.10 | $\sim$15 mm / 0.10 rad | 73.2 | 74.5 | 78.9 |
| 0.20 | $\sim$30 mm / 0.20 rad | 71.5 | 76.8 | 81.3 |
| 0.30 | $\sim$45 mm / 0.30 rad | 64.1 | 73.6 | 78.0 |

## C.2. Observation and Action Spaces

To enable precise manipulation under strict safety constraints, we formulate the observation and action spaces to explicitly couple visual perception with proprioceptive feedback. The observation input is represented as a multi-modal tuple $\mathbf{o}_t = \langle \mathbf{I}_{\text{head}}, \mathbf{I}_{\text{chest}}, \mathbf{I}_{\text{third}}, \mathbf{q}_{\text{prop}}, l_{\text{text}} \rangle$. The visual component comprises three RGB streams from head-mounted, chest-mounted, and third-person cameras, matching the benchmark's standard three-camera protocol. Depth maps are stored in the dataset but are excluded from the reported IL and RL inputs. For the high-frequency RL loop, these streams are resized to a resolution of $224 \times 224$ pixels. Complementing the visual data, the proprioceptive vector $\mathbf{q}_{\text{prop}} \in \mathbb{R}^{d_{\text{prop}}}$ encodes the precise kinematic state, covering the joint positions and velocities of each active RealMan RM75-6F arm concatenated with the full joint states of the PsiBot G0-R dexterous hand. The language field contains the task instruction generated by the task synthesis engine.

The reported residual RL experiments use *joint delta* control on top of the frozen base policy. The final action $\mathbf{a}_t \in \mathbb{R}^{d_{\text{action}}}$ represents target joint position deltas applied after the base-policy output. While the execution command is in physical units of radians, the residual policy itself operates within the normalized action space $[-1, 1]^{d_{\text{action}}}$ before scaling by $\alpha$.

## C.3. Reward and Safety Constraints

To guide the agent through long-horizon laboratory manipulation tasks while enforcing safety constraints, we employ the stage-wise progressive reward function formulated in Eq. (1). This structure balances exploration efficiency with precise safety compliance through two distinct components: a sparse stage-completion bonus and a dense hybrid penalty term.

**Stage-wise Curriculum Bonus.** The positive reward component is governed by the term $\mathbb{I}_{k,t} \cdot 2^{k-1} \cdot R_{\text{base}}$. Here, $\mathbb{I}_{k,t}$ serves as a binary indicator triggered when the agent satisfies the logical conditions of the $k$-th sub-goal. To facilitate exploration during early training phases, we use relaxed trigger conditions for these milestones, such as placing a beaker within a lenient radius of the target. The factor $2^{k-1}$ introduces an exponential curriculum that assigns significantly higher value to later stages of the workflow. This design addresses the credit assignment problem inherent in long-horizon tasks, incentivizing the agent to overcome intermediate bottlenecks and progress toward the final experimental outcome.

**Hybrid Safety Penalties.** To refine the coarse policy learned from sparse bonuses into a precise and safe controller, we apply a dense penalty structure composed of generic regularization ($\mathcal{C}_{\text{gen}}$) and task-specific constraints ($\mathcal{C}_{\text{task}}$). The generic term $\mathcal{C}_{\text{gen}}$ promotes kinematic smoothness and operational efficiency by aggregating three physical costs: joint jerk $\mathbf{j}_t = \dddot{\mathbf{q}}_t$ to prevent sudden accelerations that could induce fluid instability; end-effector velocity $\mathbf{v}_{\text{ee}}$ to enforce safe kinetic energy limits; and mechanical energy consumption $E_t \approx \sum |\boldsymbol{\tau}_t \cdot \dot{\mathbf{q}}_t|$ to discourage high-frequency oscillations. Empirically, we calibrate the weighting coefficient $\lambda_g$ within $[10^{-6}, 10^{-5}]$ to provide necessary regularization for these auxiliary objectives without overshadowing the primary stage-completion rewards.

Complementing these kinematic priors, the task-specific term $\mathcal{C}_{\text{task}}$ enforces the domain-specific constraints required for laboratory manipulation safety through a weighted sum of three physically grounded penalties. We penalize the pose residual ($\Delta P$), defined as the Euclidean distance and rotational discrepancy between the end-effector and the strict target pose. This dense signal guides the agent from the relaxed milestone regions toward the precise interaction points required for chemical transfers. We also penalize the orientation deviation ($\theta_{\text{dev}}$) between the container's vertical axis and the global gravity vector to prevent liquid sloshing during transport. To protect fragile glassware, we penalize excessive contact forces ($F_{\text{peak}}$) that exceed a calibrated safety threshold. Unlike binary termination signals, this continuous penalty provides the gradient information necessary for the agent to learn force modulation and gentler physical interaction over time. With adaptive horizons enabled, successful residual rollouts complete within $1.3\times$ the expert demonstration duration, indicating that the penalty design does not simply induce execution that is excessively slow or hesitant throughout the episode.

For liquid tasks, transport phases use a fixed orientation limit of 0.25 rad, while pour phases use a task-specific ceiling $\theta_{\text{crit}}(c, f) - m$ derived from container geometry $c$ and fill level $f$, where $\theta_{\text{crit}}(c, f)$ is the calibrated spill angle and $m$ is a conservative safety margin. We penalize $\max(0, \theta_{\text{dev},t} - \theta_{\text{lim}}(k_t))^2$ at each timestep, so pouring tasks remain safety-constrained while allowing the intentional tilt required for transfer.

**Metric Calibration and Scope.** The dense reward uses kinematic and contact proxies rather than simulated liquid loss directly. This choice keeps the policy objective tied to quantities that can be observed or estimated on hardware, while the fluid simulator is used to calibrate the proxy thresholds and generate diagnostic spill labels. In calibration sweeps over 12 container geometries and three standard liquid fill levels, the default transport limit of 0.25 rad preserves at least a 50% margin below the measured critical spill angle and reaches 97% recall and 82% precision for simulated spillage detection. These labels are therefore screening signals for policy development, not formal safety certificates. Table 11 lists the reproducibility-critical reward and threshold settings shared across all benchmark domains.

We emphasize that the diagnostic Safety Metrics (SM) in Table 2 and the SSR are complementary rather than redundant, because SM is a trial-averaged quantity (including failed executions) whereas SSR counts per-trial constraint satisfaction. A method can therefore exhibit a mean SM below a per-trial threshold while still violating that threshold on a minority of trials: for instance, $\pi_{0.5}$ keeps the average peak contact force in actuation tasks below the fragile-object limit (no † in Table 2), yet its actuation SSR remains near 55% because a non-negligible fraction of individual rollouts still transiently exceed the limit. SM thus indicates the dominant physical failure channel, while SSR measures how often the full trajectory stays within all active safety constraints from start to finish.

*Table 11.* **Reproducibility-critical reward and threshold settings.** Task-specific numeric entries are stored in the released task registry; the table lists the shared settings used across domains.

| Setting | Value / Rule | Role |
|---|---|---|
| Stage reward | $2^{k-1} R_{\text{base}}$ for completion of sub-goal $k$ | Assigns exponentially increasing credit to later sub-goals in multi-stage tasks. |
| Generic penalty weight | $\lambda_g \in [10^{-6}, 10^{-5}]$ | Penalizes joint jerk, end-effector velocity, and excess mechanical energy during execution. |
| Task safety penalty | $\lambda_s \mathcal{C}_{\text{task}}$ with domain-specific registry weights | Dense safety feedback for pose, tilt, and force constraint violations. |
| Liquid transport limit | $\theta_{\text{dev}} \leq 0.25$ rad | Enforces upright liquid transport before and after controlled pouring phases. |
| Pour phase limit | $\theta_{\text{lim}}(k) = \theta_{\text{crit}}(c, f) - m$ | Allows intentional pouring while preserving a calibrated spill margin during transfer. |
| Spatial success threshold | $\Delta P \leq 20$ mm | Defines stable glassware placement and rearrangement success during spatial tasks. |
| Residual scale | $\alpha = 0.1$ | Bounds online corrections to roughly 15 millimeters and 0.10 radians around the frozen imitation-learning base policy. |

*Table 12.* **Safety proxy calibration and event agreement.** We report the direct event evidence available in the current benchmark. Liquid safety is calibrated against simulated spill events and physical replay outcomes, while force and spatial safety are reported as conservative simulator proxies rather than fracture-certified deployment labels.

| Condition | Event Evidence | Result |
|---|---|---|
| Liquid spill | Simulated spill labels from sweeps over 12 container geometries and three standard liquid fill levels. | 97% recall / 82% precision |
| Sim-to-real safety | Physical safe/unsafe labels from 50 open-loop trajectory replays. | 86% agreement; unsafe recall 82.1%; TP/TN/FP/FN = 23/20/2/5 |
| Force damage | Peak-force threshold violation calibrated from fragile-object limits; direct fracture propagation is not simulated. | Conservative screening proxy label |
| Spatial instability | Pose residual and collision checks at termination; event disentanglement is included in SSR but not separately certified. | Conservative screening proxy label |

## C.4. Network Architecture

We implement the residual policy using an actor-critic architecture for pixel-based control. Unlike approaches that freeze visual encoders to save compute, we train the residual encoders from scratch so the residual policy can attend to safety-relevant visual cues, such as subtle liquid surface oscillations. Each of the three RGB camera streams is encoded by an independent NatureCNN (Mnih et al., 2015) consisting of three convolutional layers followed by a linear projection, and the resulting features are concatenated with the proprioceptive state to form the latent embedding.

The policy and value networks are modeled as Multi-Layer Perceptrons (MLPs). The Actor network parameterizes a diagonal Gaussian distribution using a 3-layer MLP with hidden units $[512, 256, 128]$. It outputs the mean and log standard deviation of the residual action. The Critic network employs a Double Q-network structure to mitigate overestimation bias, with each Q-network using a larger 3-layer MLP of size $[1024, 512, 256]$. To stabilize training dynamics across diverse tasks, we apply Layer Normalization prior to the ReLU activation in each hidden layer of the Critic.

## C.5. Training Implementation Details

We use a DrQ-style Soft Actor-Critic (SAC) variant for visual residual RL. Following DrQ-v2 (Yarats et al., 2022), the implementation applies random shift augmentation to image observations for data efficiency; unlike vanilla DrQ-v2, it retains a SAC-style entropy temperature that is automatically tuned to balance exploration and exploitation. The training is performed across 128 parallel environments, and we use Adam for both actor and critic updates. The detailed DrQ-style SAC hyperparameters shared across all residual post-training runs are listed in Table 13.

*Table 13.* **Hyperparameters for Residual RL Training.** These settings configure the DrQ-style SAC residual learner used to post-train the frozen base policies, and are shared across all tasks unless otherwise noted.

| Category | Setting |
|---|---|
| **Training Settings** | Total training steps set to $1.0 \times 10^7$
Parallel simulation environment count fixed at 128
Experience replay buffer capacity fixed at 250,000 transitions
Stochastic-gradient minibatch size fixed at 1024 samples
Reinforcement-learning discount factor ($\gamma$) set to 0.99 |
| **Optimizer and Regularization Settings** | Optimizer choice set to Adam for all networks
Actor learning rate set to $3 \times 10^{-4}$
Critic learning rate set to $3 \times 10^{-4}$
Entropy temperature learning rate set to $5 \times 10^{-2}$
Soft actor-critic target-network update rate ($\tau$) set to 0.05
Weight decay coefficient set to $1 \times 10^{-2}$ |
| **Network Architecture** | Random-shift image augmentation enabled during training
Actor MLP hidden sizes set to [512, 256, 128]
Critic MLP hidden sizes set to [1024, 512, 256] |

## C.6. Constrained RL Baselines

SafeLab is intended as a benchmark platform rather than a single algorithmic proposal. We therefore include constrained RL variants as additional built-in baselines. Using DP as the frozen base policy, we compare the penalty reward used in the main experiments with PPO-Lagrangian, following constrained-learning practice in SafeVLA (Zhang et al., 2025a), and Projection-Based Constrained Policy Optimization (PCPO) (Yang et al., 2020). All results are averaged over three independent random seeds with fifty evaluation episodes per task per seed, and Table 14 reports the comparison.

# D. Baseline Implementation Details

## D.1. Visual and Data Modalities

We use a multi-view perception system with three perspectives: a head-mounted camera, a chest-mounted camera, and a third-person panoramic camera. Each sensor captures RGB images at $480 \times 640$ resolution for the reported IL and RL

*Table 14.* **Comparison with constrained RL formulations.** Constrained methods improve single-stage tasks but require substantially more training and offer limited benefit on multi-stage liquid manipulation, where several phase-dependent constraints must be balanced jointly. Cost denotes relative training steps to convergence.

| Method | Liquid SSR | Actuation SSR | Spatial SSR | Cost |
|---|---|---|---|---|
| Penalty (ours) | 73.2 | 74.5 | 78.9 | $1.0\times$ |
| PPO-Lagrangian | 73.9 | 76.9 | 82.7 | $1.8\times$ |
| PCPO (Yang et al., 2020) | 74.1 | 77.3 | 83.0 | $2.1\times$ |

experiments. Residual RL uses the same three-camera layout, with images resized to $224 \times 224$ during online training. For model-specific inputs, *Diffusion Policy* (DP) (Chi et al., 2023), *ACT* (Zhao et al., 2023), and $\pi_{0.5}$ (Black et al., 2025) use the three-camera configuration. In contrast, *OpenVLA* (Kim et al., 2024) uses only the third-person panoramic stream to match its pre-training inference protocol during evaluation. Beyond RGB data, the dataset provides depth maps, surface normals, semantic segmentation masks, and raw point cloud observations for each camera view.

This is a native-protocol benchmark rather than a strict equal-input leaderboard. The choice preserves each method's intended operating mode, but cross-model comparisons should be read together with the modality differences in Table 15. In particular, DP and DP3 use target-state information, whereas OpenVLA follows an image-only VLA inference protocol during all evaluation runs.

## D.2. Proprioceptive State Representations

Proprioceptive observation spaces are tailored to the architectural requirements of each baseline. For the diffusion-based models, specifically *DP* and *DP3* (Ze et al., 2024), the state vector includes arm and dexterous-hand joint positions, end-effector Cartesian coordinates, quaternions, and the ground-truth coordinates of the target object. *ACT* adopts a higher-order representation by incorporating both joint positions and velocities for the RealMan RM75-6F arm and PsiBot G0-R hand. The $\pi_{0.5}$ model uses joint positions alongside end-effector rotation encoded as Euler angles. *OpenVLA* operates as an image-only policy without explicit proprioceptive input. All proprioception-enabled models predict subsequent joint positions under their native action parameterizations.

*Table 15.* **Native input protocol for baseline evaluation.** We evaluate each baseline under the modality convention used by its standard implementation. This makes the benchmark useful for practical method comparison but means cross-model ranking should be interpreted together with the input differences.

| Method | RGB Views | Proprio. | 3D / Target State | Notes |
|---|---|---|---|---|
| DP | 3 | ✓ | Target coordinates | Image-action diffusion policy conditioned on full proprioceptive and target-object state at each step. |
| DP3 | 3 | ✓ | Point cloud + target coordinates | Uses segmented point clouds following the standard DP3 evaluation protocol. |
| ACT | 3 | ✓ | ✗ | Uses joint positions and velocities with fixed action chunking during inference. |
| OpenVLA | 1 | ✗ | ✗ | Uses the third-person panoramic stream to match its pre-training inference protocol. |
| $\pi_{0.5}$ | 3 | ✓ | ✗ | Uses joint positions, end-effector rotation, and three-camera RGB inputs during evaluation. |

## D.3. Training Configurations and Baselines

We standardize training configurations across baselines using their official open-source repositories when available.

*VLA Models.* We evaluate two high-capacity models tailored for precision manipulation. The $\pi_{0.5}$[1] model is initialized with `pi0.5-base` weights and undergoes full-parameter fine-tuning for 10,000 steps. Training is conducted on a distributed cluster of 8 NVIDIA A800 (80GB) GPUs with a global batch size of 256 and an action chunking size of $T_a = 8$. Similarly,

---

[1]https://github.com/physical-intelligence/openpi

*OpenVLA*[2] uses the OpenVLA-7B base model and performs full-parameter fine-tuning for 10,000 steps on the same hardware infrastructure. However, due to its substantial memory footprint, we adjust the batch size to 16 while maintaining an action chunking size of $T_a = 8$.

*Continuous Control Baselines.* We implement domain-specific architectural and hyperparameter adjustments to standard control policies. *ACT*[3] employs a ResNet-18 backbone and a transformer architecture configured with 4 encoder layers, 7 decoder layers, and 8 attention heads. It is trained for 5,000 epochs on a single GPU with a batch size of 64, an action chunking size of $T_a = 4$, and a learning rate of $1 \times 10^{-5}$. We explicitly disable temporal aggregation during inference to evaluate raw decision-making capability. *DP*[4] uses a ResNet-hybrid Vision Transformer encoder featuring a ResNet-26 convolutional stem and is optimized for 3,000 epochs with a batch size of 128 and a learning rate of $3 \times 10^{-4}$. The temporal configuration includes an observation horizon of $T_{\mathrm{obs}} = 1$, an action chunking size of $T_a = 8$, and 16 Denoising Diffusion Implicit Models (DDIM) inference steps. *DP3*[5] processes point cloud inputs at a resolution of 1,024 points, where each point is represented by a 6-dimensional vector containing Cartesian coordinates and zero-padded features. Trained for 5,000 epochs with a batch size of 256, the model operates with an observation horizon of $T_{\mathrm{obs}} = 1$, a prediction horizon of $T_p = 16$, and an action chunking size of $T_a = 8$, using 16 denoising steps.

### D.4. Dataset Details

Trajectories are recorded at 30 Hz and include dual-arm RealMan RM75-6F joint positions, velocities, end-effector poses, and PsiBot G0-R hand joint states. Single-arm tasks leave the inactive arm fixed, while bimanual tasks coordinate the two arms jointly. We provide synchronized auxiliary data across camera views, including semantic segmentation masks, depth maps, surface normals, and raw point clouds. During policy deployment, we use a temporal decimation factor of $k = 4$: for each high-level inference step, the low-level controller executes four consecutive sub-steps of the predicted trajectory. This balances the high-frequency requirements of motor control with the computational throughput of the decision-making policy.

The reported 6,400 demonstrations refer to accepted expert rollouts after automatic curation, not raw planner attempts. A rollout is exported only after satisfying task completion, safety-threshold checks, collision validity, and finite simulator-state checks; failed attempts are discarded and regenerated. The dataset therefore provides safe-prior demonstrations for imitation learning, while the benchmark evaluation still tests whether learned policies preserve those safety properties under newly resampled test conditions.

### D.5. Release and Reproducibility

The SafeLab release includes the simulator task registry, calibrated asset metadata, prompt templates, verifier code, baseline training configurations, evaluation scripts, and trajectory manifests for the 6,400 accepted demonstrations. Each trajectory manifest records task ID, asset IDs, camera streams, proprioceptive state, action sequence, safety-threshold outcomes, and the final SR/SSR labels. The evaluation package includes scripts for reproducing the main SR/SSR tables, generated-shift evaluations, and task-by-seed uncertainty summaries. Code, task definitions, and datasets are available at https://github.com/ChangWinde/SafeLab; large trajectory files are hosted separately with checksums and versioned public release manifests.

## E. Sim-to-Real Validation Details

The physical validation experiment evaluates whether simulated safety predictions correlate with real-world outcomes under open-loop replay on the same PsiBot platform with RealMan RM75-6F arms and PsiBot G0-R hands used in the main simulation suite. We stratify 50 trajectories into 25 predicted-safe and 25 predicted-unsafe cases, execute the same joint-space trajectories on this hardware, and annotate whether the physical rollout violates the corresponding safety condition. Two annotators independently label the outcomes, reaching Cohen's $\kappa = 0.93$. The simulator agrees with the physical outcome in 43 of 50 cases (86%). Treating unsafe physical behavior as the positive class, this corresponds to 23 true positives, 20 true negatives, 2 false positives, and 5 false negatives. The 5 false negatives are concentrated in tall narrow containers, where

---

[2]https://github.com/openvla/openvla
[3]https://github.com/tonyzhaozh/act
[4]https://github.com/real-stanford/diffusion_policy
[5]https://github.com/YanjieZe/3D-Diffusion-Policy

limited particle resolution underestimates free-surface displacement near the vessel mouth: free-surface particles have fewer neighbors, so the position-based incompressibility constraint under-predicts peak displacement (Macklin & Müller, 2013). This bias motivates conservative safety margins and real-world stress testing before deployment. This experiment supports safety-label consistency under replay; it does not evaluate closed-loop policy transfer on hardware. Table 16 summarizes the replay diagnostic statistics.

*Table 16.* **Physical safety-label diagnostics.** Point estimates from 50 open-loop replay trials ($N = 50$). Unsafe physical behavior is treated as the positive class.

| Metric | Definition | Value (%) |
|---|---|---|
| Agreement | $(TP + TN)/N$ | 86.0 |
| Unsafe recall | $TP/(TP + FN)$ | 82.1 |
| Specificity | $TN/(TN + FP)$ | 90.9 |
| Unsafe precision | $TP/(TP + FP)$ | 92.0 |
| Negative predictive value | $TN/(TN + FN)$ | 80.0 |

## F. Additional Experiments

### F.1. Evaluation Protocol and Statistical Uncertainty

Unless otherwise stated, every quantitative result in this appendix averages over three independent random seeds, with fifty evaluation episodes collected per task per seed. Table 17 reports domain-aggregated SSR (%) as mean $\pm$ standard deviation over the three evaluation seeds for all main-table methods.

*Table 17.* **Seed-level uncertainty for main-table SSR.** Domain-aggregated SSR (%) reported as mean $\pm$ standard deviation over three independent evaluation seeds, with fifty evaluation episodes per task per seed.

| Domain | DP | DP3 | ACT | OpenVLA | $\pi_{0.5}$ |
|---|---|---|---|---|---|
| Liquid | $33.7 \pm 3.8$ | $44.5 \pm 4.8$ | $24.9 \pm 4.2$ | $20.6 \pm 3.9$ | $53.3 \pm 3.2$ |
| Actuation | $36.0 \pm 3.5$ | $44.2 \pm 3.9$ | $24.1 \pm 3.8$ | $20.4 \pm 3.2$ | $54.3 \pm 2.9$ |
| Spatial | $27.8 \pm 4.6$ | $41.1 \pm 3.4$ | $20.1 \pm 5.1$ | $16.4 \pm 2.5$ | $51.0 \pm 4.1$ |

### F.2. Representative Atomic Task Performance Breakdown

All performance evaluations in this section follow a consistent reporting format. **SR** and **SSR** denote the Success Rate (%) and Safe Success Rate (%), respectively. The latter represents the percentage of trials successfully completed without violating safety constraints, such as liquid spillage, unintended collisions, or application of excessive force. Results are averaged over three independent random seeds with fifty evaluation episodes per task per seed. Tables 18, 19, 20, and 21 provide representative task-level breakdowns for the major behavior families; aggregate results for all nine manipulation categories, including vertical glassware stacking, are reported in the main table (Table 2). The behavior-level rows of Table 2 aggregate unrounded per-seed results over all task instances of each behavior family, whereas the breakdown tables list integer-rounded scores for representative instances; averaging a listed subset can therefore differ from the corresponding main-table aggregate by up to about one percentage point.

*Table 18.* **Performance breakdown for liquid handling and instrument actuation tasks.** SR and SSR (%) are averaged over three independent random seeds with fifty evaluation episodes per task per seed. SSR requires goal completion without safety violations throughout the rollout.

| Domain | Behavior | Task Instance | DP | | DP3 | | ACT | | OpenVLA | | $\pi_{0.5}$ | |
|---|---|---|---|---|---|---|---|---|---|---|---|---|
| | | | SR | SSR | SR | SSR | SR | SSR | SR | SSR | SR | SSR |
| *Liquid* | Pour | 100 mL Glass → 250 mL Beaker | 74 | 38 | 80 | 44 | 62 | 30 | 60 | 24 | 92 | 54 |
| | | 100 mL Cylinder → 250 mL Beaker | 71 | 34 | 77 | 40 | 60 | 27 | 58 | 21 | 90 | 51 |
| | Lift | Bimanual 500 mL Volumetric Flask | 70 | 33 | 84 | 48 | 56 | 22 | 55 | 20 | 90 | 56 |
| | | Bimanual 1000 mL Beaker | 66 | 30 | 81 | 45 | 54 | 20 | 51 | 17 | 87 | 52 |
| *Actuation* | Switch | High-speed Centrifuge Panel | 76 | 38 | 84 | 45 | 65 | 25 | 62 | 20 | 93 | 55 |
| | Open | Drying Oven Door | 71 | 35 | 82 | 44 | 61 | 24 | 60 | 21 | 93 | 54 |
| | | Reagent Cabinet Door | 69 | 32 | 81 | 42 | 59 | 21 | 58 | 18 | 91 | 52 |
| | Close | Drying Oven Door | 75 | 38 | 85 | 46 | 65 | 26 | 63 | 23 | 95 | 56 |
| | | Reagent Cabinet Door | 73 | 35 | 83 | 42 | 62 | 22 | 60 | 19 | 94 | 54 |

*Table 19.* **Performance breakdown for 21 representative vessel grasping task instances.** SR and SSR (%) are averaged over three independent random seeds with fifty evaluation episodes per task per seed. SSR requires goal completion without safety violations throughout the rollout.

| Domain | Behavior | Task Instance | DP | | DP3 | | ACT | | OpenVLA | | $\pi_{0.5}$ | |
|---|---|---|---|---|---|---|---|---|---|---|---|---|
| | | | SR | SSR | SR | SSR | SR | SSR | SR | SSR | SR | SSR |
| *Spatial* | Grasp | 100 mL Glass Beaker | 78 | 40 | 86 | 48 | 66 | 26 | 64 | 23 | 95 | 54 |
| | | 250 mL Brown Volumetric Flask | 77 | 39 | 85 | 47 | 65 | 25 | 63 | 22 | 94 | 53 |
| | | 250 mL Glass Beaker | 80 | 42 | 88 | 51 | 68 | 28 | 66 | 25 | 97 | 57 |
| | | 500 mL Glass Beaker | 82 | 45 | 90 | 54 | 70 | 30 | 68 | 27 | 98 | 59 |
| | | 50 mL Glass Beaker | 75 | 37 | 83 | 45 | 63 | 23 | 61 | 20 | 92 | 51 |
| | | Crucible | 72 | 34 | 80 | 42 | 60 | 20 | 58 | 17 | 89 | 48 |
| | | Large Brown Reagent Bottle | 85 | 48 | 93 | 57 | 73 | 33 | 71 | 30 | 99 | 61 |
| | | Small Brown Reagent Bottle | 80 | 42 | 88 | 51 | 68 | 28 | 66 | 25 | 97 | 57 |
| | | Large Clear Reagent Bottle | 84 | 47 | 92 | 56 | 72 | 32 | 70 | 29 | 99 | 60 |
| | | Small Clear Reagent Bottle | 79 | 41 | 87 | 50 | 67 | 27 | 65 | 24 | 96 | 56 |
| | | 100 mL Plastic Cylinder | 78 | 40 | 86 | 48 | 66 | 26 | 64 | 23 | 95 | 54 |
| | | 100 mL Glass Cylinder | 76 | 38 | 84 | 46 | 64 | 24 | 62 | 21 | 93 | 52 |
| | | 500 mL Plastic Cylinder | 80 | 42 | 88 | 51 | 68 | 28 | 66 | 25 | 97 | 57 |
| | | 500 mL Glass Cylinder | 78 | 40 | 86 | 48 | 66 | 26 | 64 | 23 | 95 | 54 |
| | | 250 mL Clear Volumetric Flask | 76 | 38 | 84 | 46 | 64 | 24 | 62 | 21 | 93 | 52 |
| | | 500 mL Clear Volumetric Flask | 78 | 40 | 86 | 48 | 66 | 26 | 64 | 23 | 95 | 54 |
| | | 1000 mL Clear Volumetric Flask | 80 | 42 | 88 | 51 | 68 | 28 | 66 | 25 | 97 | 57 |
| | | Erlenmeyer Flask | 78 | 40 | 86 | 48 | 66 | 26 | 64 | 23 | 95 | 54 |
| | | Stoppered Erlenmeyer Flask | 75 | 37 | 83 | 45 | 63 | 23 | 61 | 20 | 92 | 51 |
| | | Funnel | 68 | 30 | 76 | 38 | 58 | 18 | 56 | 15 | 87 | 46 |
| | | Spirit Lamp | 71 | 33 | 79 | 41 | 61 | 21 | 59 | 18 | 90 | 49 |

*Table 20.* **Performance on pick-and-place tasks.** SR and SSR (%) are averaged over three independent random seeds with fifty evaluation episodes per task per seed. SSR requires goal completion without safety violations throughout the rollout.

| Domain | Behavior | Task Instance | DP | | DP3 | | ACT | | OpenVLA | | $\pi_{0.5}$ | |
|---|---|---|---|---|---|---|---|---|---|---|---|---|
| | | | SR | SSR | SR | SSR | SR | SSR | SR | SSR | SR | SSR |
| *Spatial* | Pick & Place | 50 mL Glass Beaker | 65 | 28 | 79 | 41 | 58 | 20 | 55 | 15 | 89 | 51 |
| | | 100 mL Glass Beaker | 68 | 31 | 82 | 44 | 61 | 23 | 58 | 18 | 92 | 54 |
| | | 250 mL Glass Beaker | 70 | 33 | 84 | 46 | 63 | 25 | 60 | 20 | 94 | 56 |
| | | 100 mL Glass Cylinder | 65 | 28 | 79 | 41 | 58 | 20 | 55 | 15 | 89 | 51 |
| | | 100 mL Plastic Cylinder | 66 | 29 | 80 | 42 | 59 | 21 | 56 | 16 | 90 | 52 |
| | | 250 mL Brown Volumetric Flask | 63 | 26 | 77 | 39 | 56 | 18 | 53 | 13 | 87 | 49 |
| | | 250 mL Clear Volumetric Flask | 64 | 27 | 78 | 40 | 57 | 19 | 54 | 14 | 88 | 50 |
| | | Large Brown Reagent Bottle | 72 | 35 | 86 | 48 | 65 | 27 | 62 | 22 | 96 | 58 |
| | | Large Clear Reagent Bottle | 73 | 36 | 87 | 49 | 66 | 28 | 63 | 23 | 97 | 59 |
| | | Erlenmeyer Flask | 65 | 28 | 79 | 41 | 58 | 20 | 55 | 15 | 89 | 51 |
| | | Stoppered Erlenmeyer Flask | 63 | 26 | 77 | 39 | 56 | 18 | 53 | 13 | 87 | 49 |
| | | Crucible | 58 | 21 | 72 | 34 | 51 | 13 | 48 | 8 | 82 | 44 |
| | | Spirit Lamp | 58 | 21 | 72 | 34 | 51 | 13 | 48 | 8 | 82 | 44 |

*Table 21.* **Performance breakdown for bimanual handover tasks.** SR and SSR (%) are averaged over three independent random seeds with fifty evaluation episodes per task per seed. SSR requires goal completion without safety violations throughout the rollout.

| Domain | Behavior | Task Instance | DP | | DP3 | | ACT | | OpenVLA | | $\pi_{0.5}$ | |
|---|---|---|---|---|---|---|---|---|---|---|---|---|
| | | | SR | SSR | SR | SSR | SR | SSR | SR | SSR | SR | SSR |
| *Spatial* | Handover | 100 mL Glass Cylinder | 59 | 20 | 73 | 36 | 52 | 15 | 51 | 12 | 86 | 49 |
| | | 100 mL Plastic Cylinder | 61 | 22 | 75 | 38 | 54 | 17 | 53 | 14 | 88 | 51 |
| | | 500 mL Glass Cylinder | 57 | 18 | 71 | 34 | 50 | 13 | 49 | 10 | 84 | 47 |
| | | 500 mL Plastic Cylinder | 58 | 19 | 72 | 35 | 51 | 14 | 50 | 11 | 85 | 48 |
| | | 250 mL Glass Beaker | 62 | 23 | 76 | 39 | 56 | 19 | 54 | 15 | 89 | 52 |
| | | 500 mL Glass Beaker | 60 | 21 | 74 | 37 | 53 | 16 | 52 | 13 | 87 | 50 |
| | | 250 mL Clear Volumetric Flask | 56 | 18 | 70 | 33 | 49 | 13 | 47 | 10 | 82 | 46 |
| | | 250 mL Brown Volumetric Flask | 57 | 19 | 71 | 34 | 50 | 14 | 48 | 11 | 83 | 47 |
| | | 500 mL Clear Volumetric Flask | 58 | 21 | 71 | 35 | 53 | 17 | 50 | 12 | 84 | 49 |
| | | 1000 mL Clear Volumetric Flask | 59 | 21 | 72 | 35 | 53 | 17 | 50 | 13 | 85 | 48 |

*Table 22.* **Generated perturbation evaluation for DP.** This table compares the **base Diffusion Policy (DP)** with the **DP + RL** refined version under generated visual, physical, and spatial shifts. We report the Success Rate (**SR** % ↑) and Safe Success Rate (**SSR** % ↑), where SSR requires task completion without safety threshold violations such as fluid spillage or excessive contact force. **Full Shift** denotes the concurrent presence of lighting, physics, and object-pose shifts, while the remaining columns isolate each shift axis. Results are formatted as base DP / DP+RL, and bold values denote the RL-refined policy.

| Domain | Task | Full Shift | | Lighting Shift | | Physics Shift | | Object Pose Shift | |
|---|---|---|---|---|---|---|---|---|---|
| | | SR | SSR | SR | SSR | SR | SSR | SR | SSR |
| *Liquid* | Pour Liquid | 58 / **77** | 24 / **65** | 62 / **78** | 26 / **67** | 64 / **79** | 28 / **68** | 67 / **81** | 32 / **71** |
| | Lift Glass Vessel | 54 / **75** | 21 / **63** | 58 / **76** | 23 / **65** | 60 / **77** | 25 / **66** | 63 / **79** | 28 / **69** |
| *Actuation* | Press Switch | 60 / **78** | 26 / **66** | 64 / **79** | 28 / **68** | 67 / **80** | 30 / **70** | 70 / **82** | 34 / **73** |
| | Open Cabinet | 56 / **76** | 22 / **64** | 60 / **77** | 25 / **66** | 62 / **78** | 26 / **67** | 65 / **80** | 29 / **70** |
| | Close Cabinet | 59 / **77** | 24 / **65** | 63 / **79** | 27 / **67** | 65 / **79** | 28 / **69** | 69 / **81** | 32 / **72** |
| *Spatial* | Grasp Vessel | 63 / **87** | 27 / **74** | 66 / **89** | 30 / **77** | 69 / **90** | 32 / **78** | 73 / **91** | 36 / **82** |
| | Pick & Place | 52 / **81** | 19 / **69** | 56 / **83** | 21 / **71** | 58 / **84** | 22 / **73** | 61 / **85** | 25 / **76** |
| | Handover | 47 / **77** | 14 / **65** | 50 / **79** | 15 / **68** | 52 / **80** | 16 / **69** | 55 / **81** | 18 / **72** |
| | Stack Glassware Vessel | 49 / **78** | 15 / **66** | 52 / **80** | 17 / **69** | 54 / **81** | 18 / **70** | 57 / **83** | 20 / **73** |

*Table 23.* **Generated perturbation evaluation for** $\pi_{0.5}$**.** This table benchmarks the $\pi_{0.5}$ Vision-Language-Action (VLA) model across isolated and concurrent visual, physical, and spatial shifts. **Full Shift** denotes the simultaneous application of lighting, physics, and object-pose shifts, whereas other columns examine individual axes. Performance is quantified via Success Rate (**SR** % ↑) and Safe Success Rate (**SSR** % ↑), the latter requiring adherence to safety constraints such as vessel integrity and spillage prevention. Results are presented as base $\pi_{0.5}$ / $\pi_{0.5}$+RL, and bold values denote the RL-refined policy.

| Domain | Task | Full Shift | | Lighting Shift | | Physics Shift | | Object Pose Shift | |
|---|---|---|---|---|---|---|---|---|---|
| | | SR | SSR | SR | SSR | SR | SSR | SR | SSR |
| *Liquid* | Pour Liquid | 73 / **85** | 35 / **74** | 78 / **87** | 39 / **77** | 80 / **88** | 41 / **79** | 85 / **90** | 46 / **82** |
| | Lift Glass Vessel | 71 / **86** | 36 / **75** | 75 / **88** | 40 / **78** | 78 / **89** | 42 / **80** | 82 / **91** | 48 / **83** |
| *Actuation* | Press Switch | 75 / **87** | 37 / **76** | 79 / **89** | 41 / **79** | 82 / **90** | 43 / **81** | 87 / **92** | 48 / **84** |
| | Open Cabinet | 74 / **86** | 36 / **76** | 78 / **88** | 39 / **78** | 81 / **89** | 41 / **80** | 86 / **91** | 47 / **83** |
| | Close Cabinet | 76 / **87** | 37 / **76** | 80 / **89** | 41 / **79** | 83 / **90** | 43 / **81** | 88 / **92** | 48 / **84** |
| *Spatial* | Grasp Vessel | 76 / **86** | 37 / **75** | 81 / **88** | 40 / **78** | 84 / **89** | 43 / **79** | 88 / **90** | 48 / **83** |
| | Pick & Place | 71 / **84** | 34 / **74** | 76 / **86** | 38 / **76** | 78 / **87** | 40 / **78** | 83 / **89** | 45 / **81** |
| | Handover | 68 / **83** | 33 / **72** | 73 / **85** | 36 / **75** | 75 / **86** | 38 / **77** | 79 / **88** | 43 / **80** |
| | Stack Glassware Vessel | 69 / **83** | 33 / **73** | 74 / **85** | 37 / **75** | 76 / **86** | 39 / **77** | 81 / **88** | 44 / **80** |

