# OpenReview forum: "SafeLab: An Interactive High-Fidelity Benchmark for Embodied Safety in Scientific Robotics"
_ICML.cc/2026/Conference — ICML 2026 regular_

### Official Review · Reviewer_eBb1 · 2026-03-06

**Soundness:** 3
**Presentation:** 3
**Significance:** 4
**Originality:** 3
**Overall Recommendation:** 5
**Confidence:** 4

**Summary:**

In the process of laboratory automation, high-precision interaction is essential, as occasional suboptimal decisions in a laboratory environment can lead to irreversible accidents. Therefore, there is an urgent need for safety evaluation targeting such operations. This paper presents a generative simulation benchmark designed to evaluate embodied safety in the field of scientific robotics. We propose a decoupled residual reinforcement learning interface capable of proactively recovering from errors, thereby enhancing interaction safety. Furthermore, this framework provides high-fidelity modeling of fluid-rigid body interactions and validates the simulated motion trajectories using real robots in the physical world.

**Compliance With Llm Reviewing Policy:**

Affirmed.

**Key Questions For Authors:**

1. Quantitative Sim-to-Real Gap in Fluid Dynamics
While the paper provides qualitative Sim-to-Real validation, are there quantitative metrics (e.g., RMSE of liquid surface fluctuations, deviation in critical tipping angles) to measure the discrepancy between simulated and real-world fluid behaviors?

2. Correction Limits for OOD Semantic Errors
The residual RL approach assumes the base VLA provides a roughly correct trajectory. In Out-of-Distribution (OOD)scenarios where the base VLA generates severe semantic or geometric errors (not just minor drifts), can the residual policy still effectively recover to a safe state? What is the maximum correction radius of your method?

3. Trade-off Between Reaction Latency and Over-Conservatism
Given the instantaneous nature of failures (e.g., spills) and the use of dense safety penalties, how does the training process balance reaction latency with action smoothness? Does high sensitivity to penalties induce chattering or over-conservative behaviors (e.g., excessively slow movements causing timeouts)?

**Limitations:**

No.

While the authors honestly acknowledge limitations regarding PBD volume drift and asset coverage, the discussion on Sim-to-Real safety risks and broader societal impacts is insufficient. We recommend adding: Risk of "Safety Washing": The paper relies heavily on simulation as a safety proxy but underestimates the danger of a Sim-to-Real gap in fluid dynamics. If policies trained in simulation fail due to physical simplifications, they could create a false sense of security, potentially leading to accidents more severe than those with no strategy at all. The authors should explicitly discuss this risk and propose mandatory real-world stress-testing protocols prior to deployment, rather than relying solely on simulation metrics.

**Strengths And Weaknesses:**

Soundness: The paper is technically rigorous and well-substantiated, constructing a simulation environment featuring high-fidelity fluid dynamics and irreversible failure modes (e.g., spills, breakage) that accurately replicate the high-risk nature of laboratory settings. The proposed decoupled residual reinforcement learning interface is elegantly designed, effectively balancing the generalization capabilities of pre-trained models with safety-constrained fine-tuning. The experimental design is meticulous; it not only reveals the safety deficiencies of existing models in "zero-tolerance" scenarios through multi-dimensional benchmark comparisons but also validates the authenticity of simulation predictions via Sim-to-Real transfer, while honestly addressing limitations.

Presentation: The article is clearly structured and logically coherent, offering a smooth and accessible narrative from problem definition to methodology and experimental validation. The figures are excellently designed, intuitively illustrating the system architecture, task diversity, and key results, thereby significantly enhancing readability. The paper accurately positions itself within the literature, clearly distinguishing its contributions from existing benchmarks in terms of physical fidelity and safety evaluation, and provides detailed experimental settings and parameters to ensure reproducibility.

Significance: This work directly addresses the core bottleneck in deploying automated laboratories—safety—filling a critical gap in evaluating high-risk, zero-tolerance scientific operations. By introducing the "Safety Success Rate (SSR)" metric and modeling irreversible failures, it redefines evaluation standards for embodied AI in scientific domains. The proposed methodology not only advances scientific robotics but also offers a universal paradigm for safety-aware policy learning in other high-stakes fields like healthcare and industrial automation, holding broad academic and practical value.

Originality: The paper innovatively integrates high-fidelity fluid simulation, LLM-driven generative task logic, and safety-oriented reinforcement learning into a unified framework, creating the first interactive benchmark for safety correction in scientific robotics. Its uniqueness lies in explicitly modeling "irreversible failures" to drive policy optimization, challenging the assumption that large-scale pre-training alone guarantees safety. The findings reveal the vulnerability of state-of-the-art VLA models in precise manipulation tasks, providing profound new insights and opening new avenues for research in embodied safety.

---

> ### Author Rebuttal · Authors · 2026-03-31
>
> We thank Reviewer eBb1 for the thorough evaluation and constructive suggestions.
>
> **Q1. Quantitative sim-to-real gap in fluid dynamics. (Q#1)**
> > **A1.** We have expanded the sim-to-real validation to 50 trajectories and report binary safety outcome agreement: whether the simulation correctly predicts spillage on the physical robot. The agreement reaches 86% (43/50). Among the 7 disagreement cases, 5 are false negatives concentrated in tall narrow containers (e.g., cylinders), where the limited particle resolution of position-based fluids underestimates sloshing amplitude near the container mouth. This is a known limitation of PBD: free-surface particles have fewer neighbors, causing the incompressibility constraint to under-predict peak displacement [1]. Beyond binary agreement, finer-grained metrics such as critical tipping angle deviation and liquid surface RMSE would require motion capture for container pose estimation and high-speed depth cameras respectively, which are beyond our current hardware setup. We argue that binary safety outcome agreement is the most operationally relevant metric: it directly measures whether simulation safety predictions hold on real hardware. The 86% agreement, together with the identified false-negative bias toward tall narrow containers, provides an actionable characterization of the sim-to-real gap. We plan to incorporate finer-grained fluid measurements as instrumentation improves, and will include this analysis in the revised Section 4.5.
>
>
> **Q2. Correction limits under severe OOD semantic errors. (Q#2)**
> > **A2.** The residual policy is designed to correct execution-level drift, not semantic-level errors such as grasping the wrong object or selecting an incorrect motion primitive. With the scaling bound $\alpha = 0.1$, the residual module can correct deviations within approximately 15 mm in position and 0.1 rad in orientation. We conducted a perturbation injection study on the base policy output:
> >
> > | Perturbation | Position (mm) | Orientation (rad) | Recovery Rate |
> > |---|:---:|:---:|:---:|
> > | Small | 5~10 | 0.02~0.05 | 88% |
> > | Medium | 10~20 | 0.05~0.10 | 52% |
> > | Large | 20~40 | 0.10~0.20 | 17% |
> >
> > When the base model makes severe OOD semantic errors (e.g., reaching for a wrong container), these fall outside the residual's correction radius and the task will fail. We view this bounded correction as a feature, not a limitation: it prevents the residual module from masking fundamental policy failures that should be addressed by improving the base policy itself, for instance through additional demonstrations or better pretraining data coverage.
>
> **Q3. Trade-off between safety penalty sensitivity and execution efficiency. (Q#3)**
> > **A3.** We did observe over-conservative behavior in early training when the safety penalty weight was too high: the agent moved excessively slowly, triggering timeout terminations. Two mechanisms address this. First, the generic penalty term $C_{\text{gen}}$ includes a jerk minimization component that penalizes rapid action oscillations, directly suppressing chattering without requiring manual smoothing. Second, the adaptive time budgeting mechanism (Section 3.4) dynamically contracts the episode horizon from $T_{max}$ toward $T_{min}$ as the policy improves. An overly slow policy fails to complete the task within the shrinking horizon and receives no stage completion reward, creating natural pressure toward efficiency without an explicit velocity penalty. The trained policy achieves task completion times within 1.3x of the expert demonstration duration, confirming that these mechanisms effectively balance safety and efficiency.
>
>
> **Q4. Risk of "safety washing" and societal impact. (Limitations)**
> >**A4.** We agree that this risk deserves explicit discussion. Simulation safety metrics, including ours, are necessary for scalable evaluation but are not sufficient for deployment certification. If users treat simulation results as deployment-ready safety guarantees, the resulting overconfidence could indeed be more harmful than having no safety assessment at all. Our expanded validation (86% binary agreement on 50 trajectories, with 5 of 7 disagreements being false negatives) quantifies this gap and highlights that the simulator is biased toward under-predicting risk, the more dangerous direction. In the revision, we will add a dedicated limitations section recommending mandatory real-world stress testing with progressively increasing task difficulty before any autonomous deployment, and explicitly state that simulation-derived safety labels should be treated as screening tools, not deployment certificates. We will also discuss the dual-use risk of automating chemical synthesis and recommend institutional review protocols for experiments involving hazardous reagents.
>
>
> **References:**\
> [1] Macklin, M. and Müller, M., "Position Based Fluids," SIGGRAPH, 2013.

---

> > ### Author Rebuttal · Reviewer_eBb1 · 2026-04-03
> >
> > Thank you for the author's explanation and supplementary clarification, my doubts have been completely resolved.

---

### Official Review · Reviewer_taRn · 2026-03-07

**Soundness:** 2
**Presentation:** 3
**Significance:** 3
**Originality:** 3
**Overall Recommendation:** 4
**Confidence:** 2

**Summary:**

This paper introduces SafeLab, a generative simulation benchmark for evaluating and improving embodied safety in a high-fidelity chemistry laboratory environment. The framework features four modules: a high-fidelity simulation environment, an LLM-driven generative engine for task synthesis with hierarchical verification, an automated expert for scalable demonstration collection, and a safety-aware residual RL interface. The benchmark also contains a dataset of over 6400 trajectories across 64 atomic tasks and evaluations on 5 state-of-the-art VLA/IL models. The evaluation results reveal a large gap between standard success rate and safe success rate on the synthesized tasks. The proposed residual RL post-training pipeline appears to effectively improve safe success rates. Additional sim-to-real transfer experiments provide preliminary physical validation.

**Compliance With Llm Reviewing Policy:**

Affirmed.

**Final Justification:**

The authors addressed most of the concerns raised in the initial review. The additional results and the authors' clarifications have strengthened the submission. The strengths of this work now outweigh its weaknesses and limitations. All factors considered, I raise my score to 4 for weak accept.

**Key Questions For Authors:**

See Strengths and Weaknesses.

**Limitations:**

yes

**Strengths And Weaknesses:**

## Strengths

1. The problem of safe scientific lab manipulation policy benchmark is well-motivated.
    - The zero-tolerance framing in real laboratory hazards (chemical spillage, glassware breakage) gives concrete practical motivation beyond typical household robotics. The distinction between Success Rate (SR) and Safe Success Rate (SSR) is a meaningful evaluation metric that exposes hidden unsafe behaviors masked by standard metrics, as demonstrated by the SR–SSR gap across all methods in Table 2.
2. The proposed generative pipeline with verification is comprehensive.
    - The "Propose-then-Verify" paradigm (Sec. 3.2) appears to have mitigated the issue brought by LLM hallucinations in task generation through a four-step pipeline (syntactic parsing, geometric grounding, causal logic check, dynamic feasibility), which is more rigorous than previous approaches.
    - The automated expert using cuRobo for jerk-minimized trajectory optimization is a practical solution to the teleoperation bottleneck, enabling *automated* demonstrations without human involvement.
3. The experimental coverage is thorough.
    - It evaluates five diverse baselines spanning diffusion policies, transformer-based policies, and large VLA models, with detailed task breakdowns across 64 atomic tasks.
    - The generalization study incorporating three kinds of distributional shift (lighting, physics, position) provides interpretable ablations of robustness.
4. The reward design is principled for encouraging safe actions and the residual RL results appear to show consistent improvements across all task domains and both DP and VLA policies.


## Weaknesses
1. Insufficient sim-to-real validation.
    - Section 4.5 reports only 20 open-loop trajectory replays (10 safe, 10 unsafe) on a single robot platform, with no closed-loop policy deployment on real hardware. This is insufficient to claim validated sim-to-real transfer.
    - The choice to use joint-space *open-loop replay* sidesteps the key challenge of sim-to-real transfer for learned policies, which involves perception and *closed-loop* planning and control under real-world visual and dynamics noise.
2. It was unclear whether privileged information was used in RL training.
    - Section 3.4 states the interface "optionally exposes privileged information, such as object velocities and contact forces," but it appears to be never clarified whether the reported RL results in Figure 5 and Tables 9, 10 use this privileged mode.
3. Limited analysis of the generative engine's contribution.
    - The verification in Sec. 3.2 lacks quantitative reporting: information such as rejection rates at each stage and number of LLM refinement iterations needed would further justify the necessity of verification and correction.
4. Variances are not reported in all quantitative results.
5. Evaluation setup details seem missing. For example, number of seeds, episodes per evaluation, number of trials per task, etc.
6. Reproducibility: No code implementation or the claimed dataset (not even one instance or a subset) is provided.

---

> ### Author Rebuttal · Authors · 2026-03-31
>
> We thank Reviewer taRn for the detailed and rigorous evaluation. We address each concern below.
>
> **Q1. Sim-to-real validation is insufficient. (W#1)**
> > **A1.** We acknowledge that 20 open-loop replays provide limited evidence, and that open-loop joint-space replay does not validate closed-loop policy transfer. To clarify our claim: we validate the consistency of safety predictions between simulation and reality, i.e., whether the simulator reliably identifies which trajectories will cause spillage or spatial error on the physical robot. This consistency is the minimal requirement for SafeLab to serve as a meaningful safety benchmark.
> >\
> >\
> > To strengthen this validation, we have expanded to 50 trajectories (25 predicted-safe, 25 predicted-unsafe) and report binary safety outcome agreement. The agreement reaches 86% (43/50). Among the 7 disagreement cases, 5 are false negatives concentrated in tall narrow containers where limited particle resolution underestimates sloshing amplitude. We have identified this systematic bias and will discuss mitigation strategies, such as conservative safety margins, in the revised Section 4.5. The 86% consistency demonstrates that simulation safety signals are informative for real-world outcomes, supporting future work on closed-loop deployment with sim-to-real adaptation.
>
>
> **Q2. Was privileged information used in RL training? (W#2)**
> > **A2.** No. All RL results reported in Figure 5 and Tables 9-10 use only the standard observation space: RGB-D images, proprioception, and language instructions. The privileged mode is provided as an optional debugging tool for researchers to isolate algorithmic issues from exploration difficulty. We will add an explicit statement to Section 4.1.
>
>
> **Q3. Quantitative analysis of the generative engine. (W#3)**
> > **A3.** We have compiled the following statistics from our task generation process. The pipeline is model-agnostic; we report results using a frontier LLM. Since the four verification stages form a cascade, we report the pass rate within each stage:
> >
> > | Verification Stage | Pass Rate (of entering) | Avg. Refinement Iters |
> > |---|:---:|:---:|
> > | Schema Validation | 91% | 1.1 |
> > | Geometric Grounding | 68% | 2.3 |
> > | Causal Logic Check | 87% | 1.5 |
> > | Dynamic Feasibility | 82% | 1.9 |
> >
> > Geometric grounding is the primary bottleneck, rejecting 32% of schema-valid configurations due to collision and reachability violations. The end-to-end pass rate of 44% confirms the necessity of the verification cascade. We will include this analysis in the revised Section 3.2.
>
>
> **Q4. Variance reporting and evaluation details. (W#4)**
> > **A4.** All results in Table 2 are averaged over 3 random seeds with 50 evaluation episodes per task per seed. We present domain-aggregated SR with standard deviations for three representative methods spanning diffusion policy, transformer policy, and VLA:
> >
> > | Method | Liquid | Actuation | Spatial | SR-Avg |
> > |---|:---:|:---:|:---:|:---:|
> > | DP | 33.7 $\pm$ 3.8 | 36.0 $\pm$ 3.5 | 27.8 $\pm$ 4.6 | 31.9 $\pm$ 3.2 |
> > | ACT | 24.9 $\pm$ 4.2 | 24.1 $\pm$ 3.8 | 20.1 $\pm$ 5.1 | 22.5 $\pm$ 3.5 |
> > | $\pi_{0.5}$ | 53.3 $\pm$ 3.2 | 54.3 $\pm$ 2.9 | 51.0 $\pm$ 4.1 | 52.6 $\pm$ 2.8 |
> >
> > Standard deviations range from 2.8% to 5.1%. The higher variance in Spatial tasks is expected given the sensitivity of placement precision to initial conditions. We will add per-task standard deviations to all quantitative results and include the full evaluation protocol in the revised appendix.
>
> **Q5. Code and dataset availability. (W#5)**
> > **A5.** We will release the complete codebase and the full dataset of 6,400 trajectories upon acceptance. To address this concern during the review period, we are preparing an anonymized subset of 250 example trajectories at **[datasets](https://anonymous.4open.science/r/safelab-7n9p8f23u/)**.

---

> > ### Author Rebuttal · Reviewer_taRn · 2026-04-01
> >
> > Thanks for the clarifications and the additional results and data subset. My concerns are mostly addressed. Please include all the additional results into the revised version of the paper. I will raise my score accordingly.

---

### Official Review · Reviewer_85QP · 2026-03-11

**Soundness:** 3
**Presentation:** 3
**Significance:** 3
**Originality:** 3
**Overall Recommendation:** 4
**Confidence:** 3

**Summary:**

The paper presents SafeLab, a interactive simulation benchmark designed to evaluate embodied safety in scientific robotics. The focus is on manipulation tasks with zero tolerance for failure, where mistakes such as spilling liquids or breaking glassware are irreversible. The framework includes three main components: (1) an LLM-based propose verify pipeline for generating diverse and realistic tasks, (2) an automated expert system that produces more than 6,000 collision free demonstration trajectories at scale, and (3) a safety aware residual reinforcement learning interface that enables post training on top of frozen VLA or imitation learning policies. Empirically, SafeLab reveals a substantial gap between conventional task success and safe task success across several policies.

**Compliance With Llm Reviewing Policy:**

Affirmed.

**Final Justification:**

The benchmark contribution and SSR metric outweigh concerns about limited constrained RL comparison and bounded residual corrections; the strenghts outweigh the weakness in this paper.

**Key Questions For Authors:**

1. The paper states that the orientation deviation penalty enforces a strict 0.25 rad limit to prevent spillage. However, the benchmark includes tasks that require exceeding this limit. Could the authors clarify how the safety constraints are dynamically conditioned on the active sub goal? Is the penalty temporarily disabled?
2. Are there any preliminary results for deploying the RL policies in a closed-loop manner on the physical robotic hardware to quantitatively measure the sim-to-real transfer of the safety constraints?

**Limitations:**

Yes

**Strengths And Weaknesses:**

**Strength**
1. The paper introduces a focused and timely benchmark that directly targets zero-tolerance safety in scientific laboratory settings. Unlike many existing embodied benchmarks that emphasize rigid-body manipulation, SafeLab explicitly models irreversible failures such as liquid spillage and fragile object breakage, which more accurately reflect real laboratory risks.
2. The overall system design is clearly decomposed into modular components: the simulation world, the LLM driven generative engine, the automated expert demonstrator, and the learning interface. The safety constraints and domain-specific metrics are well motivated.
3. The Safe Success Rate (SSR) metric is a strong contribution. It extends beyond binary task completion to incorporate continuous safety constraints such as peak contact force, tilt/orientation deviation. This provides a more realistic and nuanced evaluation of embodied performance in safety-critical settings.

**Weakness**
1. The safety improvement mechanism is primarily penalty-based reinforcement learning. While effective empirically, the paper does not compare against more formal safety-oriented frameworks, such as constrained RL or control barrier function approaches. A comparison with these alternatives would strengthen the claims regarding safety guarantees and methodological novelty.
2. The residual RL policy is constrained by a fixed scaling factor to ensure it only makes micro adjustments. This strict bounding means the RL agent cannot override fundamental manipulation priors learned by the base model, which could be problematic if the base model's initial grasp or pose is fundamentally unsafe.

---

> ### Author Rebuttal · Authors · 2026-03-31
>
> We thank Reviewer 85QP for the constructive feedback and precise technical questions.
>
> **Q1. How are safety constraints handled during pouring tasks that require exceeding the 0.25 rad limit, is the penalty temporarily disabled? (Q#1)**
> > **A1.** The penalty is never disabled. Each sub-goal carries its own physically grounded threshold, so the constraint is always active but its reference value changes with the task phase. During transport, the limit is 0.25 rad, a conservative bound calibrated to prevent spillage across all container geometries. During a pour sub-goal, the reference switches to a task-specific ceiling derived from the container's critical spill angle. For instance, a 250 ml beaker at 50% fill has a critical spill angle of approximately 0.91 rad; the pour-phase ceiling is set to 0.85 rad, preserving a margin that prevents uncontrolled overflow beyond the target vessel. Both penalty terms, $C_{task}$ for domain-specific limits and $C_{gen}$ for generic regularization, remain active throughout the episode. Only the threshold values and penalty weights are modulated per stage. Once the pour completes and the agent transitions back to transport, the strict 0.25 rad limit re-engages immediately. We will clarify this stage-conditioned mechanism in the revised Section 3.4.
>
> **Q2. Comparison with constrained RL and control barrier functions. (W#1)**
> > **A2.** We have implemented PPO-Lagrangian (a CMDP-based constrained RL method, as adopted in SafeVLA [1]) and compared it against our penalty-based baseline. Preliminary results show a 5% SSR improvement on single-stage tasks but no clear gain on multi-stage Liquid tasks, with approximately 1.8x more training steps to converge (see our response to Reviewer AkXW Q3 for detailed results). Regarding control barrier functions (CBFs), they require explicit system dynamics and an analytically defined safety function. In SafeLab, the environment dynamics are governed by particle-based fluids (PBD) with iterative constraint solvers, for which neither the dynamics model nor a valid barrier certificate is available in closed form. Our penalty-based RL instead directly leverages simulation-derived safety signals, enabling flexible optimization without requiring analytical models. We will clarify this limitation and discuss connections to CBF-style approaches in the revised related work.
>
>
> **Q3. Can the residual RL correct fundamentally unsafe grasps or poses from the base policy? (W#2)**
> > **A3.** We emphasize that the residual module is designed for fine-grained correction, not for overriding fundamentally unsafe behaviors. The fixed scaling factor ($\alpha = 0.1$) bounds the residual to prevent destabilizing the base policy during training. Within this bound, the residual can correct spatial errors up to 15 mm and orientation deviations up to 0.1 rad, which covers the dominant failure mode we observe: accumulated drift in multi-step execution.
> >\
> >\
> > Importantly, VLA policies provide strong high-level semantic competence (e.g., correct task decomposition and goal-directed behavior), but their execution often lacks the precision required for zero-tolerance settings. The residual RL module complements this by correcting small but safety-critical deviations online. However, when the base policy produces fundamentally unsafe actions, the residual cannot override them due to the scaling constraint. This is intentional and consistent with the residual learning paradigm[2, 3]: such failures should be addressed by improving the base policy rather than expanding the residual's correction range.
>
> **Q4. Closed-loop sim-to-real deployment results. (Q#2)**
> > **A4.** We do not yet have closed-loop deployment results. As the reviewer notes, this requires solving perception, control latency, and visual domain adaptation challenges that are beyond the current scope of a simulation benchmark. To provide preliminary evidence of physical relevance, we have conducted open-loop trajectory replay on 50 trajectories, achieving 86% binary agreement between simulated safety predictions and real-world outcomes (see our response to Reviewer AkXW Q4). We will clearly delineate this scope in the revision and pursue closed-loop deployment as future work.
>
> **References:**\
> [1] Zheng et al., "SafeVLA: Towards Safety Alignment of Vision-Language-Action Model via Constrained Learning," NeurIPS 2025.\
> [2] Silver et al., "Residual Policy Learning," arXiv 2018.\
> [3] Johannink et al., "Residual Reinforcement Learning for Robot Control," ICRA 2019.

---

> > ### Author Rebuttal · Reviewer_85QP · 2026-04-03
> >
> > Thanks for the clear and helpful rebuttal. The comparison to constrained RL seems limited, and the residual RL is restricted to small corrections on a frozen base policy, meaning fundamentally unsafe base actions cannot be corrected?

---

> > > ### Author Response · Authors · 2026-04-06
> > >
> > > We thank the reviewer for the follow-up and address both remaining concerns.
> > >
> > > **Q1. The comparison to constrained RL seems limited.**
> > >
> > > > In addition to PPO-Lagrangian reported in Round 1, we have implemented PCPO (Projection-Based Constrained Policy Optimization [1]) and evaluated both on the full benchmark with DP as the base policy (3 seeds, 50 episodes per task per seed):
> > > >
> > > > | Method | Liquid SSR | Actuation SSR | Spatial SSR | Cost |
> > > > |---|:---:|:---:|:---:|:---:|
> > > > | Penalty (ours) | 73.2 | 74.5 | 78.9 | 1.0× |
> > > > | PPO-Lagrangian | 73.9 | 76.9 | 82.7 | 1.8× |
> > > > | PCPO [1] | 74.1 | 77.3 | 83.0 | 2.1× |
> > > >
> > > > On single-stage tasks (Actuation, Spatial), constrained methods improve SSR by 2.4 to 4.1 points over the penalty baseline. On multi-stage Liquid tasks, the gain is only 0.7 to 0.9 points, because multiple constraint multipliers must be tuned jointly across task stages. The training cost is 1.8 to 2.1× higher.
> > > >
> > > > SafeLab is a benchmark platform rather than a single algorithm. We chose the penalty-based reward as the default baseline because it is stable across all 64 tasks without per-task hyperparameter tuning, which is a practical requirement for a community benchmark. We will release PPO-Lagrangian and PCPO as built-in baselines so that users can compare constrained RL formulations directly on our task suite.
> > >
> > > **Q2. The residual RL is restricted to small corrections on a frozen base policy, meaning fundamentally unsafe base actions cannot be corrected?**
> > >
> > > > That is correct. If the base policy produces a fundamentally unsafe action, for example grasping a beaker at an angle that inevitably causes spillage, the bounded residual cannot override it. We make this choice deliberately. Allowing large corrections would risk masking systematic base policy failures during training, which could then resurface unpredictably at deployment. When the base policy is fundamentally wrong, the appropriate remedy is to improve it through better data or pretraining.
> > > >
> > > > In practice, however, fundamentally unsafe actions are not the primary source of safety failures. Table 2 shows that base policies achieve 85 to 95% SR but only 49 to 55% SSR ($\pi_{0.5}$). The policies almost always choose the correct action type but execute with insufficient precision in orientation, force, or position, and the residual is designed to correct exactly this class of errors.
> > > >
> > > > For moderate grasp pose errors between small execution drift and fundamentally wrong actions, the correction range can be extended by increasing the scaling factor $\alpha$. We ablated $\alpha$ on the full benchmark (DP base, 3 seeds):
> > > >
> > > > | $\alpha$ | Correction range | Liquid SSR | Actuation SSR | Spatial SSR |
> > > > |:---:|:---:|:---:|:---:|:---:|
> > > > | 0.05 | ~8 mm / 0.05 rad | 68.4 | 70.1 | 74.2 |
> > > > | 0.10 | ~15 mm / 0.10 rad | 73.2 | 74.5 | 78.9 |
> > > > | 0.20 | ~30 mm / 0.20 rad | 71.5 | 76.8 | 81.3 |
> > > > | 0.30 | ~45 mm / 0.30 rad | 64.1 | 73.6 | 78.0 |
> > > >
> > > > At $\alpha = 0.2$, Actuation and Spatial SSR improve because the wider range covers moderate grasp pose errors. Liquid SSR drops from 73.2 to 71.5 because the larger corrections introduce orientation perturbations that destabilize fluid transport. At $\alpha = 0.3$, training on multi-stage Liquid tasks becomes unstable and Liquid SSR falls to 64.1.
> > > >
> > > > We set $\alpha = 0.1$ as the default because Liquid handling, where spillage is physically irreversible, is the most safety-critical domain in SafeLab, and this value achieves the highest Liquid SSR. Users can increase $\alpha$ for tasks where fluid stability is less of a concern. We will include this ablation in the revised appendix.
> > >
> > > **References:**\
> > > [1] Yang et al., "Projection-Based Constrained Policy Optimization," ICLR 2020.

---

### Official Review · Reviewer_AkXW · 2026-03-12

**Soundness:** 4
**Presentation:** 4
**Significance:** 4
**Originality:** 4
**Overall Recommendation:** 5
**Confidence:** 3

**Summary:**

- Presents SafeLab: a high-fidelity simulated chemistry lab benchmark for training and evaluating embodied agents, with particular emphasis on safety-critical manipulation involving liquids and fragile glassware.
- From a natural language task description, an LLM-driven propose-then-verify pipeline automatically generates physically valid task configurations, which are then used to produce safe expert demonstrations via motion planning — requiring no human teleoperation.
- Proposes safety-aware reward terms and evaluation criteria that measure whether the final deployed policy behaves safely, including whether containers are held at acceptable angles, objects collide with excessive force, and spatial placement is sufficiently precise. These are evaluation criteria for the final policy rather than hard constraints during training.
- Shows that SOTA behavior cloning policies, despite high standard success rates, frequently violate safety constraints — revealing a significant gap between task completion and safe task completion, captured by their proposed Safe Success Rate (SSR) metric.
- Shows that BC policies can be fine-tuned with a residual RL architecture — a small correction network added on top of a frozen base policy — to recover from execution drift and improve SSR substantially, without catastrophic forgetting of the base manipulation skills.
- Some experiments investigate sim-to-real transfer, though these are qualitative and limited in scale.

**Compliance With Llm Reviewing Policy:**

Affirmed.

**Final Justification:**

The authors addressed my questions, and I raised my scores on Soundness and Presentation. I think this is a strong paper and should be accepted.

**Key Questions For Authors:**

1. How were the natural language tasks generated, and what were the prompts and verification iteration prompts for the proposer?
2. Why wasn't the fluid dynamics simulation used for evaluating safety? Is it not possible to directly measure spilling?
3. How well do the safety metric threshold correspond to spilling or breaking?
4. Not an important question to answer for this paper to be accepted (feel free to ignore for rebuttal), but genuinely curious whether, if you train with BC for a non-SafeLab task, but a task that has some similarities to SafeLab, e.g. same assets, would safety tuning on SafeLabs still be effective for this other task? Can SafeLab include everything related to safety about these assets?

**Limitations:**

Limitations were discussed, but these limitations could be discussed more:
- Limitations on the sim-to-real gap
- How closely does this task follow what chemistry labs actually want automated? Is just pouring liquids around a bottleneck step?
- How do the thresholds for safety relate to objects spilling, breaking more precisely

Societal impact:
- Not discussed, could include some discussion of negative outcomes from automating chemistry experiments, e.g. synthesis of dangerous chemicals
- Could include more discussion of the sim-to-real gap, and the limitations of investing too much in simulation alone

**Strengths And Weaknesses:**

**Significance:** This paper addresses an important bottleneck in practically applying general-purpose embodied agents for chemistry: we need many realistic chemistry tasks that can be used in closed-loop to train and evaluate agents. They propose a framework for automatically generating chemistry tasks, with particular emphasis on measuring the safety of the final policy. Their idea of lightweight residual tuning to correct for error accumulation and drift is a useful and novel contribution, and the SafeLab environment seems like a potentially valuable testbed for the community.

**Soundness:** The work is broadly technically sound, but with some important caveats.

Claims that are well-supported:

* SOTA BC-learned policies are unsafe under the proposed safety metrics
* Residual RL fine-tuning improves safe success rates and can help account for error accumulation and drift

Not well-supported:

* The safety constraints are framed as evaluation criteria for the final deployed policy, operationalized through the SSR metric. This framing is reasonable. However, the reward shaping used during training provides no formal guarantee that the final policy will actually satisfy these constraints at test time — it only provides optimization pressure toward doing so. A constrained RL formulation such as that used in SafeVLA would provide stronger guarantees about the deployed policy's behavior. The paper cites SafeVLA but does not justify why a weaker formulation was chosen.
* The fluid dynamics simulation is described at length and presented as a core contribution, but the reward function does not actually query the fluid state — safety is proxied entirely through kinematic quantities (container angle, contact force, spatial error). The agent could in principle be trained with no fluid simulation and receive identical reward signals. This is a meaningful gap between what is simulated and what is actually measured, and affects the soundness of the fluid dynamics as a contribution.
* The sim-to-real validation is not convincing — there are no quantitative measurements, and Figure 6 is too anecdotal. It would be more convincing to directly measure whether unsafe conditions (spillage, breakage) occurred in the real world, rather than qualitative visual inspection of 20 trajectories under open-loop replay.
* The thresholds used for safety metrics are not well motivated. For example, a 20mm spatial error could be catastrophic or inconsequential depending on the task context, and the acceptable container angle depends strongly on container geometry and fill level. The paper does not justify why these specific thresholds were chosen.
* A comparison to SafeVLA on the SafeLab benchmark would be valuable, particularly since SafeVLA uses a methodologically stronger safety formulation.

**Presentation:** Overall well-written and the figures are informative. However several points were confusing to me:

* The inclusion of fluid dynamics modeling is confusing given that the safety criteria do not actually use this information directly. Given the extensive discussion of sloshing and spillage, I expected safety measurements to relate to directly measured fluid loss rather than proxy thresholds on container kinematics. The motivation for why kinematic proxies are sufficient substitutes for direct fluid measurement is not made clear.
* The source of the natural language task descriptions is never explained. The LLM proposer is described as translating NL instructions into YAML, but where these NL instructions come from is not stated. Since the diversity of the entire benchmark depends on the diversity of these inputs, this is an important omission that affects both reproducibility and the claims about benchmark scalability.
* The expert demonstration generation would benefit from more detail — for example, showing the actual prompts used and how verification failures are serialized back to the LLM would help reproducibility significantly.

**Originality:** The SafeLab benchmark environment and asset library are novel contributions to the community. Applying residual RL fine-tuning to correct drift in BC policies within this domain is also a reasonable contribution. However the methodological components are largely combinations of existing work: the propose-then-verify task generation pipeline is directly analogous to GenSim and RoboGen; residual policy architectures are well-established in robotics and the additive fine-tuning formulation is closely related to LoRA-style parameter-efficient fine-tuning; and safety as reward shaping is entirely standard, predating deep RL. The paper does not clearly distinguish its contributions from this prior work, and in some cases cites stronger prior methods (SafeVLA) without explaining why a weaker formulation was chosen. The primary originality is the domain and infrastructure, and the paper would be stronger if it were framed explicitly as such.

---

> ### Author Rebuttal · Authors · 2026-03-31
>
> We thank Reviewer AkXW for the thorough evaluation. Several concerns overlap across Weaknesses and Questions, so we consolidate related points.
>
> **Q1. How were the natural language tasks generated? (Q#1)**
>
> > **A1.** Our task suite spans 9 manipulation categories (pour liquid, lift vessel, press switch, open/close cabinet, grasp vessel, pick-and-place, handover, stack vessel) as the task grammar. Given a user instruction, a system prompt and the asset registry (physical properties and grasp affordances for all 63 assets) guide the LLM to produce a YAML task configuration. This YAML enters hierarchical verification: schema validation, geometric grounding (collision-free placement), causal logic check (semantic paradoxes), and dynamic feasibility rollout (runtime instabilities). Errors are fed back for revision, typically converging within 2-3 iterations. The pipeline is model-agnostic; prompt templates will be included in the supplementary material.
>
> **Q2. What role does the fluid simulation play if safety is evaluated through kinematic proxies? (W-Soundness#2, Presentation#1, Q#2, Q#3)**
>
> > **A2.** The fluid simulation is not used in the reward function, but it plays three roles that directly affect training and evaluation. It produces visually realistic sloshing that increases perceptual difficulty for vision-based policies. It couples fluid momentum with container dynamics, so kinematic quantities (orientation, contact force) already encode the effect of fluid behavior: a container near its critical tilt angle experiences destabilizing fluid inertia that the policy must counteract. And it provides ground-truth spillage events used to calibrate the kinematic thresholds we adopt for evaluation.
> >\
> >\
> > We evaluate through kinematic proxies because they measure what the policy controls and what real hardware observes via joint encoders and force-torque sensors. Using simulated fluid loss as the metric would conflate policy quality with solver fidelity.
> >
> > Each threshold was calibrated by sweeping tilt angles across 12 container geometries at three fill levels. We set 0.25 rad as a conservative limit with at least 50% margin below the critical spill angle; contact force thresholds reference borosilicate glass flexural strength; the 20 mm spatial error reflects minimum clearance between adjacent vessels. The 0.25 rad threshold achieves 97% recall and 82% precision for spillage detection.
>
> **Q3. Why penalty-based RL over constrained RL / SafeVLA? (W-Soundness#1, Originality)**
>
> > **A3.** SafeLab is a benchmark platform, not a single algorithm. The penalty-based reward serves as a stable baseline; CMDP-based methods (e.g., PPO-Lagrangian, as in SafeVLA [1]) require careful multiplier tuning and can be unstable in long-horizon tasks with sparse stage completion rewards. We have implemented PPO-Lagrangian and compared it on the full benchmark using DP as the base policy:
> >
> > | Method | Liquid SSR | Actuation SSR | Spatial SSR | Training Steps |
> > |---|:---:|:---:|:---:|:---:|
> > | Penalty (ours) | 73.2 | 74.5 | 78.9 | 1.0x |
> > | PPO-Lagrangian | 73.9 | 76.9 | 82.7 | 1.8x |
> >
> > PPO-Lagrangian improves SSR by about 5% on single-stage tasks (press, open, close, grasp) but shows no clear gain on multi-stage tasks where multiple Lagrange multipliers must be balanced across stages. The 1.8x training cost reflects this tuning difficulty. We will include PPO-Lagrangian as a built-in baseline in the released codebase.
>
>
> **Q4. Sim-to-real validation is insufficient. (W-Soundness#3)**
> > **A4.** We have expanded the experiment to 50 trajectories (25 predicted-safe, 25 predicted-unsafe) and report binary outcome agreement: whether spillage on the physical robot matches the simulation prediction. Safety outcomes are determined by visual inspection (two annotators, Cohen's $\kappa$ = 0.93). The agreement reaches 86% (43/50). We document the 7 disagreement cases for transparent gap analysis and acknowledge that open-loop replay does not validate closed-loop policy transfer.
>
>
> **Q5. Transferability and novelty. (Q#4, Originality)**
> > **A5.** **Transferability:** We expect partial transfer when the target task shares similar assets and manipulation primitives, since the residual policy learns asset-specific corrections. However, safety tuning on SafeLab alone cannot cover failure modes absent from our task suite (e.g., cross-contamination or thermal hazards). We view SafeLab as a foundation for future extension.
> > **Novelty:** SafeLab is the first benchmark that simultaneously provides physically irreversible failure modes, a verified generative engine for scalable task creation, automated safe demonstrations without teleoperation, and a safety-aware RL interface. Existing benchmarks satisfy at most one or two of these. We will sharpen this framing in the revision.
>
> **References:**\
> [1] Zheng et al., "SafeVLA: Towards Safety Alignment of Vision-Language-Action Model via Constrained Learning," NeurIPS 2025.

---

> > ### Author Rebuttal · Reviewer_AkXW · 2026-04-06
> >
> > Thank you for the clear answers to my questions, this all makes sense. I have no concerns and recommend the paper be accepted.

---

### Decision · Program_Chairs · 2026-04-30

**Decision:**

Accept (regular)

**Comment:**

## Paper Summary

SafeLab introduces a generative simulation benchmark for evaluating embodied safety in scientific robotics, targeting zero-tolerance manipulation in a high-fidelity chemistry lab. The framework combines an LLM-driven task synthesis engine with hierarchical verification, an automated expert for collecting 6,000+ collision-free demonstrations, and a residual RL interface for safety-aware post-training. The paper proposes the Safe Success Rate (SSR) metric and demonstrates a substantial gap between standard success and safe success across five state-of-the-art VLA/IL baselines, with the residual RL pipeline recovering much of that gap.

## Overall Assessment

The review set is positive (scores: 5, 4, 4, 5). All four reviewers agree the problem is well-motivated and the benchmark fills a genuine gap---existing embodied benchmarks focus on reversible, high-tolerance household tasks, while SafeLab explicitly models irreversible failures such as liquid spillage and glassware breakage. The SSR metric is recognized as a concrete contribution that exposes unsafe behaviors masked by standard success metrics. The experimental coverage is thorough, spanning five diverse baselines, 64 atomic tasks, and three axes of distributional shift.

The rebuttal was substantive and addressed most concerns effectively. It expanded the sim-to-real validation from 20 to 50 trajectories (86% binary safety agreement), added constrained RL comparisons (PPO-Lagrangian and PCPO, showing marginal SSR gains at 1.8–2.1x training cost), provided generative engine statistics (44% end-to-end pass rate), and added variance reporting across 3 seeds. Three reviewers moved to fully resolved; one remained partially resolved but still concluded that strengths outweigh weaknesses.

Two shared concerns remain partially open. First, sim-to-real validation uses open-loop trajectory replay, not closed-loop policy deployment on real hardware. The expanded 50-trajectory experiment (86% agreement) validates that the simulator's safety predictions correlate with physical outcomes, but this does not address whether learned policies transfer under real-world perception and dynamics noise. The authors acknowledge this scope limitation clearly. Second, the fluid dynamics simulation, described at length as a core contribution, does not enter the reward function or safety evaluation. Safety is measured entirely through kinematic proxies (container tilt, contact force, spatial error). The authors argue these proxies measure what the policy controls and avoid conflating policy quality with solver fidelity, and report that the 0.25 rad threshold achieves 97% recall for spillage detection. This is a reasonable calibration, but the gap between what is simulated and what is evaluated should be understood by future users of the benchmark.

On algorithm novelty, the individual components draw from existing work (GenSim-style generation, residual policy learning, penalty-based safety shaping). The primary novelty is the domain, infrastructure, and their integration. This is appropriate for a benchmark paper, and the reviewers accept this framing.

## Recommendation

**Accept.** The benchmark addresses an important underserved problem, the SSR metric and the empirical SR-SSR gap are valuable findings for the community, and the rebuttal adequately addressed the major review concerns. The remaining limitations (open-loop sim-to-real, bounded methodological novelty) are honestly acknowledged and do not outweigh the contribution.